# Dual-Stream Diffusion for World-Model Augmented Vision-Language-Action Model

John Won [1]   Kyungmin Lee [1]   Huiwon Jang [1 2]   Dongyoung Kim [1 2]   Jinwoo Shin [1 2]

## Abstract

Augmenting Vision-Language-Action models (VLAs) with world models is promising for robotic policy learning but faces challenges in jointly predicting states and actions due to the modality gap. To address this, we propose DUal-STream diffusion (DUST), a world-model augmented VLA framework featuring a multimodal diffusion transformer that maintains separate modality streams while enabling cross-modal knowledge sharing. In addition, DUST utilizes independent noise perturbations and a decoupled flow matching loss to learn cross-modal causal relationships. We further introduce an asynchronous sampling method for action and vision tokens that enhances performance through inference-time scaling. Experimental results on simulated benchmarks like RoboCasa and GR-1 show that DUST achieves up to 6% gains over state-of-the-art VLA and world-modeling baselines, with inference-time scaling providing an additional 2–5% improvement. In real-world tasks using the Franka Research 3, DUST outperforms baselines by 10% in success rate. Finally, we demonstrate that DUST enables effective transfer learning through both pretraining on action-free videos and joint-training with heterogeneous robot and human datasets. Project page here.

## 1. Introduction

Vision-Language-Action models (VLAs) have emerged as a powerful paradigm for general-purpose robotic policies (Black et al., 2025; NVIDIA et al., 2025; Brohan et al., 2023; Li et al., 2023b; Kim et al., 2024; Luo et al., 2025; Shukor

et al., 2025), building upon the rich perceptual grounding of internet-scale vision-language models (VLMs). By integrating action experts on top of existing VLM models (*e.g.*, diffusion policy (Chi et al., 2023)), VLAs generate precise, instruction-conditioned actions that generalize across novel objects, scenes, and instructions (Zawalski et al., 2024). However, despite their strong perceptual grounding and instruction following capabilities, these models often lack an explicit grasp of the underlying physical processes, struggling to model how their actions will ultimately transform the environment (Guo et al., 2024).

To bridge this gap, recent research has augmented VLAs with world-modeling objectives that jointly predict future observations and actions (Guo et al., 2024; Zheng et al., 2025; Liang et al., 2025). Learning the joint distribution of the two modalities enables the models to effectively capture the latent dynamics that govern both actions and their visual results, improving performance and generalization. Previous approaches typically rely on unified joint diffusion (*e.g.*, see Figure 1a), which forces modalities into a single latent space (Guo et al., 2024; Huang et al., 2025). This often results in a mismatch between predicting low-dimensional, temporally smooth action trajectories and high-dimensional, spatially complex visual data. In contrast, causal designs that separate these modalities into distinct models (*e.g.*, see Figure 1b) usually implement uni-directional information flow by conditioning action generation on world modeling outputs (Liang et al., 2025; Hu et al., 2025). This approach helps with learning modality-specific structures, yet inherently prevents bidirectional knowledge transfer. As such, designing world-models and action prediction models remains a challenge due to the trade-off between cross-modal integration and modality-specific fidelity.

To bridge these contrasting approaches, we propose **du**al-**st**ream Diffusion (**DUST**) to resolve this trade-off by maintaining distinct modality streams while facilitating cross-modal exchange (see Figure 1c). Built on a multimodal diffusion transformer (MMDiT) (Esser et al., 2024), DUST utilizes separate token streams for actions and observations that interact through shared cross-modal attention layers. On top of this architecture, we introduce a decoupled diffusion training algorithm that applies independent noising sched-

[1]Kim Jaechul Graduate School of AI, Korea Advanced Institute of Technology, Seoul, Republic of Korea [2]RLWRLD, Seoul, Republic of Korea. Correspondence to: Jinwoo Shin <jinwoos@kaist.ac.kr>.

*Proceedings of the 43$^{rd}$ International Conference on Machine Learning*, Seoul, South Korea. PMLR 306, 2026. Copyright 2026 by the author(s).

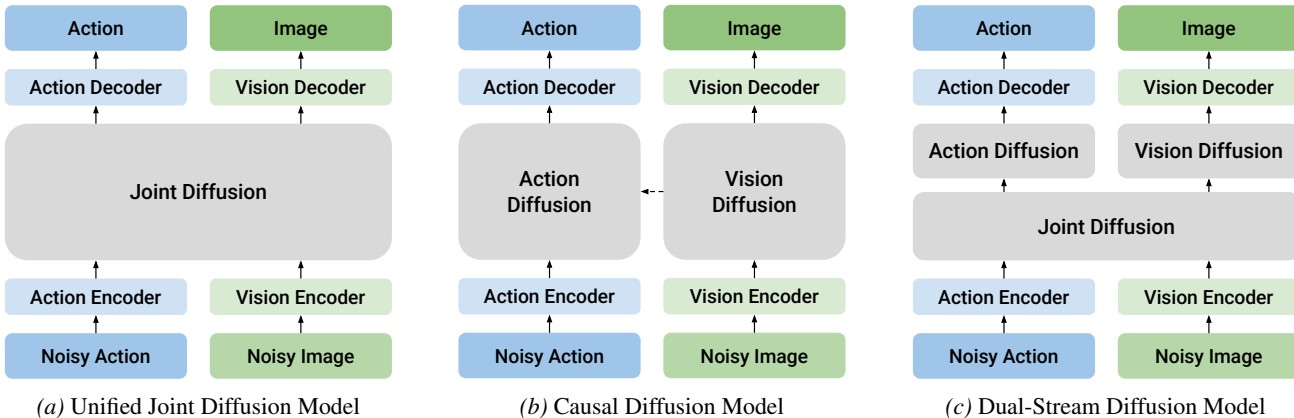

*(a)* Unified Joint Diffusion Model     *(b)* Causal Diffusion Model     *(c)* Dual-Stream Diffusion Model

*Figure 1.* **Architectures of world-model augmented VLAs.** (a) Unified Joint Diffusion concatenates action and vision tokens and generates both with a single model. (b) Causal Diffusion uses separate models with one-way conditioning. (c) **Dual-Stream Diffusion (ours)** maintains separate streams for each modality while enabling cross-modal knowledge transfer through shared attention.

ules to each modality, enabling the model to learn causal relationships between them under various noise configurations (Chen et al., 2025a; Rojas et al., 2025). The network is optimized via modality-specific flow matching losses, allowing actions and observations to evolve according to their respective statistical structures. Finally, we introduce a sampling strategy for DUST that jointly samples action and visual observations. Specifically, in order to handle the difference between the modalities, we introduce asynchronous denoising, where we take diffusion steps on the high-dimensional vision tokens more frequently than the low-dimensional action tokens. As a result, our approach allows test-time scaling that balances efficiency and accuracy.

We evaluate the effectiveness and scalability of DUST through extensive evaluations on simulated, real-world, and transfer learning scenarios. To analyze DUST's pretraining performance, we freeze the backbone VLM (Li et al., 2025b) and train the diffusion-based action expert (Chi et al., 2023) across all experiments. In simulation benchmarks, DUST outperforms baselines such as standard VLAs (*e.g.*, GR00T-N1.5 (NVIDIA et al., 2025)) and world-modeling approaches (*e.g.*, FLARE (Zheng et al., 2025)) on the Robo-Casa and GR-1 benchmarks, achieving success rate gains of 5% and 6%, respectively. The asynchronous joint sampling strategy also proves effective at test-time, providing an additional 2–6% boost over naive sampling approaches. When evaluating on a Franka Research 3 arm, DUST achieves the highest success rates across a diverse suite of tasks, outperforming baselines by more than 10%, and demonstrates robust real-world performance and physically consistent predictions across environments. Lastly, we leverage DUST in pretraining with action-free videos and show that DUST exhibits significant gains when transferred to downstream tasks. We also demonstrate its capacity for joint training on diverse robot-human mixtures, providing a unified framework for scaling VLA models with heterogeneous data.

## 2. Related Works

**Vision-language-action models (VLAs).** VLAs have recently emerged as a promising paradigm for general-purpose robot policy learning, extending vision–language models (VLMs) pretrained on internet-scale multimodal datasets (Dai et al., 2023; Team, 2024; Xiao et al., 2024), VLA architectures adapt them for robotics by either generating actions autoregressively (Kim et al., 2024; Brohan et al., 2023; Wu et al., 2024; Cheang et al., 2024) or employing diffusion-based action experts (Black et al., 2025; NVIDIA et al., 2025). In this work, we adopt the diffusion modeling formulation for action generation. Beyond these designs, recent extensions explore cross-embodiment latent action spaces (Ye et al., 2025; Bu et al., 2025b) and reasoning-driven architectures for complex task execution (Zawalski et al., 2024). Despite these advances, most approaches emphasize imitation-based action distribution learning without explicitly modeling how actions influence future states. In contrast, our framework integrates a world-modeling objective that captures physical dynamics, enabling more grounded and effective action generation.

**World-modeling for robotic policy learning.** Prior work has augmented VLAs with world-modeling objectives that generate future states alongside action generation. One line of research, exemplified by PAD (Guo et al., 2024) and EnerVerse (Huang et al., 2025), employs unified architectures that jointly model future images and actions through diffusion (Figure 1a). UWM (Zhu et al., 2025) extends this approach with modality-specific time schedules, while FLARE (Zheng et al., 2025) introduces implicit world-modeling by aligning mid-level features to future image embeddings instead of directly diffusing them. UVA (Li et al., 2025a) embeds both modalities into a shared latent space, followed by modality-specific decoders that reconstruct the original data. A complementary line of work, including Video Pol-

icy (Liang et al., 2025) and Video Prediction Policy (Hu et al., 2025), adopts disjoint architectures that allow only unidirectional conditioning between modalities (Figure 1b). Ranasinghe et al. (2026) employ a sequential visual-then-action denoising pipeline where predicted future optical flow guides action generation.

Concurrent to our work, a complementary line of *world action models* (WAMs) have emerged, which repurpose large video generation backbones as joint video-and-action predictors, including Cosmos Policy (Kim et al., 2026), Fast-WAM (Yuan et al., 2026), and DreamZero (Ye et al., 2026). These approaches are largely orthogonal to ours: whereas WAMs derive policies from video-generation priors, our framework augments a VLA with world-modeling objectives to improve action generation directly.

Another key design choice concerns how future states are represented. A common approach used in PAD (Guo et al., 2024), PIDM (Tian et al., 2025), and This&That (Wang et al., 2024) is to directly reconstruct the future RGB observation. In contrast, methods such as DINO-WM (Zhou et al., 2024) and FLARE (Zheng et al., 2025) replace RGB prediction with the generation of future observation embeddings derived from pretrained encoders like DINO-V2 (Oquab et al., 2024) and Q-Former (Li et al., 2023a). We adopt this embedding-based strategy, as it emphasizes the semantic structure of future states while avoiding the need to reproduce pixel-level details, which is information that is often irrelevant for downstream control, yet costly to model.

## 3. Preliminaries

**Problem setup.** Let $\mathcal{D} = \{T_1, T_2, ...\}$ be the dataset composed of expert demonstration trajectories, where each trajectory $T_i = \{I, \{(O_t, A_t)\}_{t=0}^L\}$ consists of task instruction $I$ and observations $O_t$ and action sequences $A_t$. Specifically, we denote the observations at timestep $t$ as $O_t = (o_t^v, o_t^s)$, where $o_t^v$ is the visual observation and $o_t^s$ is the robot proprioceptive state. Actions are grouped in chunks (Zhao et al., 2023; Chi et al., 2023) such that $A_t = (a_t, ..., a_{t+k-1})$ where $k$ is the chunk length. Our goal is to train a model that predicts $A_t$, given $O_t$ and $I$.

**Vision-language-action model (VLA).** In developing a VLA model to solve this problem, we follow common practice introduced in recent diffusion-based VLA models (Black et al., 2025; NVIDIA et al., 2025). Specifically, we use a pretrained vision-language model (VLM; Li et al. 2025b) to extract high-level semantic information from the image observations and text instruction. Then, the extracted representations are used as conditions for the action expert through cross-attention layers in a diffusion transformer (DiT; Peebles & Xie 2022) during action prediction.

The action expert is optimized using the flow matching

objective (Lipman et al., 2023). Formally, given an action sequence $A_t$, we sample a random timestep $\tau \in [0, 1]$ and Gaussian noise $\epsilon \sim \mathcal{N}(\mathbf{0}, \mathbf{I})$ to construct noisy action $A_t^\tau = \tau A_t + (1 - \tau)\epsilon$. Let $\Phi_t$ denote the visual-language features extracted from the VLM, conditioned on the current visual observation $o_t^v$ and instruction $I$. The velocity network $V_\theta(\Phi_t, A_t^\tau, o_t^s)$ is trained to predict the ground-truth velocity field $A_t - \epsilon$ with the following flow matching loss:

$$\mathcal{L}_{\text{FM}}(\theta) = \mathbb{E}_{A_t^\tau, \tau}\left[\left\|V_\theta(\Phi_t, A_t^\tau, o_t^s) - (A_t - \epsilon)\right\|^2\right], \quad (1)$$

where we sample timestep $\tau$ from a beta distribution as $\tau \sim \text{Beta}(\frac{s-\tau}{s}; 1.5, 1.0)$ with $s = 0.999$ following common practice (Black et al., 2025; NVIDIA et al., 2025). During inference, we initialize the action sequence with Gaussian noise as $A_t^0 \sim \mathcal{N}(\mathbf{0}, \mathbf{I})$, and integrate the learned velocity field using Euler's method to generate action chunks over $N_A$ denoising steps with stride $\Delta\tau = 1/N_A$:

$$A_t^{\tau+\Delta\tau} = A_t^\tau + V_\theta(\Phi_t, A_t^\tau, o_t^s)\Delta\tau. \quad (2)$$

**World-modeling.** A fundamental limitation of standard VLA models is that they learn a direct mapping from observations to actions without explicitly reasoning about how those actions will affect the environment. World-modeling addresses this by augmenting the policy with a vision predictive objective. Concretely, let $o_{t+k}^v$ denote the visual observation obtained after executing the action chunk $A_t$ of length $k$. A naïve approach would train the model to directly reconstruct $o_{t+k}^v$ in pixel space, as done in prior work (Guo et al., 2024; Tian et al., 2025; Wang et al., 2024). However, pixel-level prediction forces the model to allocate capacity toward reproducing high-frequency visual details such as textures, lighting variations, and background clutter that carry little information for downstream control. This not only increases the computational burden but can actively hinder the learning of physically meaningful dynamics, as the loss landscape becomes dominated by perceptually salient but control-irrelevant details.

To circumvent this issue, we instead operate in a learned embedding space, following recent approaches that have demonstrated the effectiveness of predicting future states in the representation space of pretrained encoders (Zhou et al., 2024; Zheng et al., 2025). We define the world-modeling target $\tilde{o}_{t+k}$ as the representation of the future observation $o_{t+k}^v$ produced by the vision encoder of our VLM backbone. These embeddings capture the semantic and structural content of the scene, while abstracting away low-level visual noise. The world-modeling objective is then to predict $\tilde{o}_{t+k}$ conditioned on the VLM features $\Phi_t$ and the proprioceptive state $o_t^s$. This way, the model focuses on learning the causal structure of the environment, specifically on how actions transform the scene at a level of abstraction that directly supports effective policy learning.

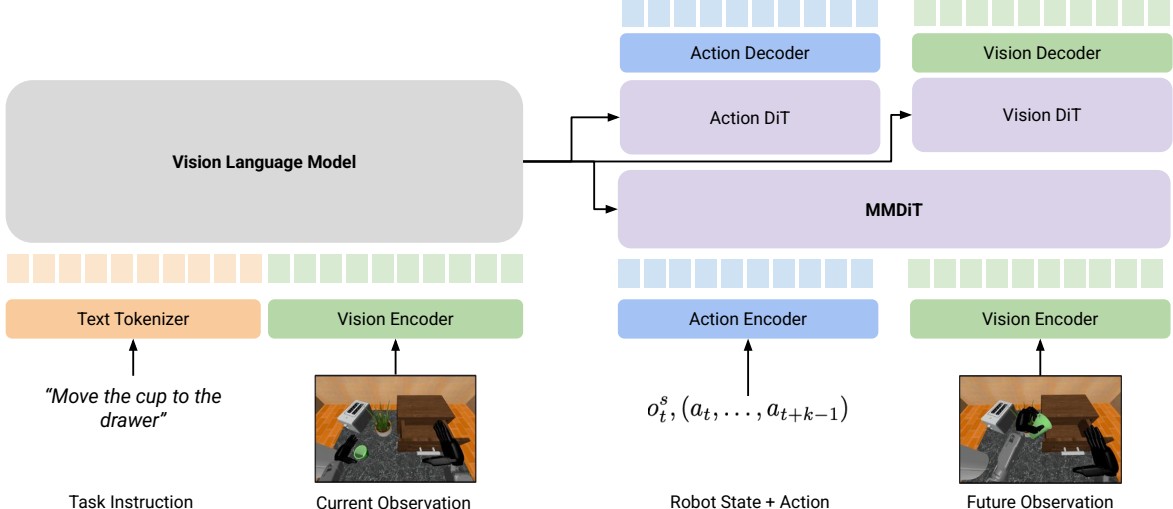

*Figure 2.* **Dual-stream diffusion (DUST) architecture.** Our architecture has a **(1)** VLM model VLM$\phi(\cdot)$ that processes current observation and task instruction to produce semantic representations, and a **(2)** diffusion model $\pi_\theta$ which conditions on these representations to generate actions and future observation embeddings.

## 4. Method

In this section, we present the **du**al-**st**ream diffusion (DUST) model, our framework designed for joint world-modeling and action prediction. The core challenge we address is the inherent conflict within joint modeling of the two modalities, actions and future observations, which have fundamentally different statistical properties. Our method systematically resolves this conflict through three key contributions. We first introduce the DUST architecture (Section 4.1), which utilizes a multimodal diffusion transformer to maintain modality-specific pathways while enabling cross-modal information exchange. We then detail our decoupled training algorithm (Section 4.2), which employs independent noise levels for each modality during training to optimize a joint objective. Finally, we describe a novel joint sampling strategy (Section 4.3) that supports test-time scaling by evolving the two modalities at different rates.

### 4.1. DUST architecture

To effectively model both action trajectories and future image observations, our architecture must strike a balance between specialized processing and cross-modal integration. As illustrated in Figure 2, DUST is built upon a central vision-language model (VLM) backbone that provides semantic features $\Phi_t$ from the current observation and task instruction. This conditioning is fed into our core diffusion model $\pi_\theta$, which takes the triplet $(o_t^s, A_t^\tau, \tilde{o}_{t+k}^\tau)$ as input, which is composed of the robot state, noised actions, and noised future observation embeddings.

This input is processed by a stack of multimodal diffusion transformer (MMDiT) blocks. Critically, within each MMDiT block, the action and vision token streams are prop-

agated through separate pathways. They are concatenated only during the shared cross-modal attention layer, which facilitates information exchange, and are split back into their respective streams for all other operations. To further decouple their dynamics and directly support our training objective (described in Section 4.2), each stream receives its own distinct timestep embedding via adaptive layernorm (AdaLN) (Peebles & Xie, 2022). After traversing the shared MMDiT layers, the two streams are routed into modality-specific DiT blocks for specialized denoising. This final stage allows the vision pathway to focus on reconstructing semantically consistent future embeddings, while the action pathway refines the low-level motor control, thereby improving the joint modeling of control and world dynamics.

### 4.2. Joint training algorithm

We now introduce a joint training algorithm based on a decoupled diffusion framework. Our design is inspired by diffusion forcing (Chen et al., 2025a), which utilizes independent per-token noise, and we adapt this to a per-modality scheme, where actions and future image embeddings are noised independently using timesteps $\tau_A$ and $\tau_o$. This decoupling breaks the synchronized corruption of standard diffusion, forcing the model to learn bidirectional causal dependencies. For instance, denoising with a clean future observation and a fully noised action ($\tau_o \approx 0, \tau_A \approx T$) supervises the inverse dynamics: "What action leads to this state?" Conversely, predicting a noised observation from a clean action supervises the forward dynamics: "What is the consequence of this action?" By exposing the model to these varied noise combinations, we ensure it captures the underlying causal structure of the task, facilitating both robust planning and accurate future prediction.

**Decoupled noise scheduling.** Let the two modalities be the actions $A_t \in \mathbb{R}^{k \times d_A}$ and the future observation embedding $\tilde{o}_{t+k} \in \mathbb{R}^{d_o}$, where $d_A$ and $d_o$ are the dimensions of the actions and image embedder, respectively. During training, we sample timesteps independently, with $\tau_A \in [0,1]$ for actions and $\tau_o \in [0,1]$ for future vision. Let $\epsilon_A, \epsilon_o \sim \mathcal{N}(\mathbf{0}, \mathbf{I})$ be sampled Gaussian noise, with which we noise $A_t$ and $\tilde{o}_{t+k}$, giving the noisy action sequences and noisy future observations as $A_t^{\tau_A} = \tau_A A_t + (1 - \tau_A)\epsilon_A$ and $\tilde{o}_{t+k}^{\tau_o} = \tau_o \tilde{o}_{t+k} + (1 - \tau_o)\epsilon_o$, respectively. The diffusion model $V_\theta$ predicts the velocity field of each modality, conditioned on the VLM feature $\Phi_t$. Let us denote $V_\theta(\Phi_t, A_t^{\tau_A}, \tilde{o}_{t+k}^{\tau_o}, o_t^s) = [V_\theta^A, V_\theta^o]$ as the outputs of diffusion model. Then, the training objective for actions and vision (*i.e.*, world-modeling) are given as follows:

$$\mathcal{L}_{\mathrm{A}}(\theta) = \mathbb{E}_{A_t^{\tau_A}, \tilde{o}_{t+k}^{\tau_o}}\left[\left\|V_\theta^A - (A_t - \epsilon_A)\right\|^2\right],$$
$$\mathcal{L}_{\mathrm{WM}}(\theta) = \mathbb{E}_{A_t^{\tau_A}, \tilde{o}_{t+k}^{\tau_o}}\left[\left\|V_\theta^o - (\tilde{o}_{t+k} - \epsilon_o)\right\|^2\right]. \quad (3)$$

To effectively train the model over this joint objective, we adopt the results of Rojas et al. (2025), which demonstrate that we can decompose the joint objective of diffusing two modalities into the sum of unimodal diffusion losses, given that we utilize independent noise injection for each modality. Concretely, we can utilize the following loss:

$$\mathcal{L}_{\mathrm{Joint}}(\theta) = \mathcal{L}_{\mathrm{A}}(\theta) + \lambda_{\mathrm{WM}}\mathcal{L}_{\mathrm{WM}}(\theta), \quad (4)$$

where $\lambda_{\mathrm{WM}} > 0$ is a weighting hyperparameter.

### 4.3. Vision-action joint sampling

During inference, we jointly sample actions and vision in parallel, by leveraging bidirectional dependencies whereby generated actions constrain plausible future states, and predicted states guide action generation. However, the two modalities are not symmetric in their requirements: image embedding diffusion operates in a high-dimensional space and typically benefits from many denoising steps, whereas low-dimensional action diffusion often converges in far fewer steps and even loses performance when sampled over many steps. To address this disparity and exploit our decoupled design, we introduce a test-time scaling strategy based on asynchronous forward Euler sampling.

In this scheme, we first sample action noise $A_t^0 \sim \mathcal{N}(0, I_A)$ and future vision noise $\tilde{o}_{t+k}^0 \sim \mathcal{N}(0, I_v)$. We define a fixed number of diffusion steps for actions, $N_A$, and a potentially larger number of steps for vision, $N_o = q \times N_A$, where $q \in \mathbb{N}$. The sampling process then proceeds using a global timestep $\Delta \tau_o = 1/N_o$. As shown in Figure 3, the vision tokens are updated at every single fine-grained step. In contrast, the action tokens are updated only every $q$ steps, corresponding to their larger step size $\Delta \tau_A = 1/N_A =$

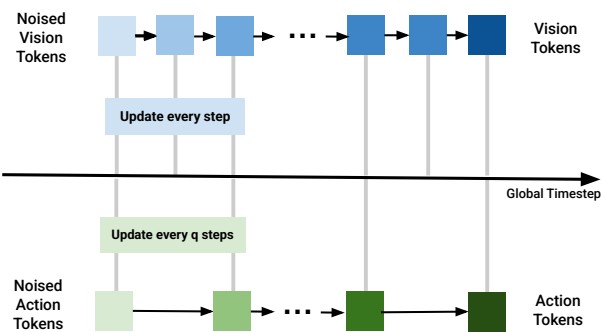

*Figure 3.* **Vision-action joint sampling.** During inference, we sample over $N_A$ steps for action tokens and $N_o = q \times N_A$ steps for vision tokens. The global timestep advances by $\Delta\tau_o = 1/N_o$, where vision tokens are updated every step and action tokens are updated only every $q$ steps in $\Delta\tau_A = 1/N_A$ strides. The default $q$ value is 1, and increasing it allows test-time scaling.

$q\Delta\tau_o$. This asynchronous integration is defined as:

$$\tilde{o}_{t+k}^{\tau_o + \Delta\tau_o} = \tilde{o}_{t+k}^{\tau_o} + V_\theta^o\,\Delta\tau_o,$$

$$A_t^{\tau_A + \Delta\tau_A} = \begin{cases} A_t^{\tau_A} + V_\theta^A\,\Delta\tau_A, & \text{if } (\tau_A N_o \bmod q = 0), \\ A_t^{\tau_A}, & \text{otherwise.} \end{cases}$$
$$(5)$$

For our main experiments, we use $q = 1$ (setting $N_o = N_A = 4$) for a fair comparison with baselines. In Section 5.3, we explore the benefits of this test-time scaling by increasing $q$ (and thus $N_o$), creating a tunable trade-off between inference speed and predictive accuracy.

## 5. Experiments

In this section, we empirically assess the effectiveness of DUST. Section 5.1 presents results from simulated environments (RoboCasa, GR-1) and real-world (Franka Research 3) tasks. In Section 5.2, we investigate transferability with pretraining and joint training settings. In Section 5.3, we assess the effectiveness of our joint sampling method for test-time scaling. In Section 5.4, we analyze the various components of our methodology. We also present results from CALVIN and LIBERO in Appendix A.1.

**VLM backbone and diffusion architecture.** We adopt the Eagle-2 model (Li et al., 2025b) as our frozen VLM backbone to process image observations and task instructions. Semantic features are extracted from the 12th layer of the VLM and used as conditioning signals for the action expert. The diffusion backbone consists of 12 MMDiT blocks for joint-modal processing, followed by 4 modality-specific DiT blocks for both streams. Conditioning with VLM features is applied in an interleaved manner, with alternating self-attention and cross-attention layers.

*Table 1.* **Evaluation on RoboCasa.** Success rates (%) on RoboCasa for 8 pick-and-place (PnP), 6 contraption open/close (OP/CL), and 10 other miscellaneous tasks. 100, 300, and 1,000 demos per task are used for training. [†]: reproduced results.

| Method | 100 Demos | | | | 300 Demos | | | | 1,000 Demos | | | |
|---|---|---|---|---|---|---|---|---|---|---|---|---|
| | PnP | OP/CL | Other | Avg. | PnP | OP/CL | Other | Avg. | PnP | OP/CL | Other | Avg. |
| PAD | 11.3 | 42.9 | 34.6 | 26.7 | 14.3 | 53.8 | 42.4 | 33.2 | - | - | - | - |
| VPP | 16.0 | 62.9 | 46.2 | 37.1 | 21.3 | 72.5 | 53.2 | 43.7 | - | - | - | - |
| $\pi_0$-FAST | 0.7 | 13.7 | 22.6 | 13.1 | 1.5 | 48.7 | 31.0 | 25.6 | 12.0 | 35.7 | 36.6 | 28.2 |
| $\pi_0$ | 16.8 | 67.7 | 49.2 | 43.0 | 15.0 | 73.3 | 49.4 | 43.9 | 13.0 | 69.7 | 58.0 | 45.9 |
| GR00T-N1.5 | 21.5 | 60.3 | 46.8 | 41.7 | 27.2 | 66.0 | 46.6 | 45.0 | 32.3 | 75.7 | 50.8 | 50.8 |
| + FLARE[†] | 23.0 | 64.8 | 49.8 | 44.6 | 38.0 | 76.7 | 56.2 | 55.3 | 45.9 | 83.7 | 68.2 | 64.6 |
| **+ DUST** | **29.5** | **76.0** | **51.0** | **50.1** | **42.3** | **80.7** | **58.1** | **58.5** | **48.3** | **86.3** | **68.6** | **66.3** |

*Table 2.* **Evaluation on GR-1.** Success rates (%) on GR-1 for 18 pick-and-place (PnP) and 6 articulated (Art.) tasks. 300 and 1,000 demos per task used. [†]: reproduced results.

| Method | 300 Demos | | | 1,000 Demos | | |
|---|---|---|---|---|---|---|
| | PnP | Art. | Avg. | PnP | Art. | Avg. |
| PAD | 11.2 | 15.0 | 12.2 | - | - | - |
| VPP | 20.8 | 18.7 | 20.2 | - | - | - |
| $\pi_0$-FAST | 16.8 | 29.4 | 20.0 | 19.2 | 30.8 | 22.1 |
| $\pi_0$ | 19.4 | 32.3 | 22.7 | 20.8 | 34.3 | 24.2 |
| GR00T-N1.5 | 17.6 | 28.3 | 20.3 | 30.7 | 31.0 | 30.8 |
| + FLARE[†] | 34.0 | 33.0 | 33.7 | 39.3 | 32.4 | 36.3 |
| **+ DUST** | **35.8** | **36.7** | **36.0** | **42.2** | **41.3** | **42.0** |

*Table 3.* **Evaluation on real-world tasks.** Success rates (%) of 4 pick-and-place (PnP) tasks, 1 insertion task, and 2 tool-using tasks for real-world Franka Research 3 robot experiments. [†]: reproduced results.

| Method | Pick-and-Place | | | | Insert | Tool-Use | | |
|---|---|---|---|---|---|---|---|---|
| | PnP-1 | PnP-2 | PnP-3 | PnP-4 | Cord | Eraser | Brush | Avg. |
| $\pi_0$ | 50.0 | 64.6 | 45.8 | 33.3 | 8.3 | 37.5 | 41.7 | 40.2 |
| GR00T-N1.5 | 58.3 | 75.0 | 50.0 | 35.4 | 12.5 | 50.0 | 44.4 | 46.5 |
| + FLARE[†] | 62.5 | 72.9 | 50.0 | 37.5 | 20.8 | 54.2 | 48.6 | 49.5 |
| **+ DUST** | **83.3** | **79.2** | **62.5** | **45.8** | **29.2** | **56.3** | **65.3** | **59.9** |

**World-modeling target.** We follow recent works in avoiding direct pixel-level prediction. The world-modeling target $\tilde{o}_{t+k}$ is the future image embedding derived from the SIGLIP-2 (Tschannen et al., 2025) representations produced by the Eagle-2 model, providing a rich, semantic target for the vision stream to predict. Each image yields 256 tokens from the embedding model, which are reduced to 64 tokens via $2 \times 2$ average pooling. In total, the diffusion module processes 1 state token, 16 action tokens, and 64 future image tokens. For our joint loss (Eq. 4), we set $\lambda_{WM} = 1.0$, equally weighting the action and world-modeling objectives based on our ablation study (Section 5.4).

**Baselines.** Our primary baselines are the vanilla GR00T-N1.5 model (NVIDIA et al., 2025), representing the current state-of-the-art in VLA models, and a variant trained with FLARE loss (Zheng et al., 2025). Because the FLARE implementation has not been released, we reimplemented it to match DUST as closely as possible, using the same VLM backbone and the same world-modeling objective (see Section A.3 for details). To ensure fair comparison, all models based on GR00T-N1.5 are trained with a frozen pretrained VLM module and a randomly initialized diffusion action-expert module. We also train and evaluate the $\pi_0$ (Black et al., 2025) and $\pi_0$-FAST (Pertsch et al., 2025) models, initializing both from the PaliGemma VLM backbone (Beyer et al., 2024) rather than from a robot-pretrained VLA checkpoint. Finally, for the 100-/300-demo RoboCasa settings and the 300-demo GR-1 setting, we further include PAD (Guo et al., 2024) and Video Prediction Policy (VPP (Hu et al., 2025)), enabling comparison against other explicit world-modeling policies.

### 5.1. Main results

First, we verify the efficacy of DUST across 2 simulated environments and 1 real-world setting. For the simulated setting, we utilize the RoboCasa (Nasiriany et al., 2024) and GR-1 (NVIDIA et al., 2025) benchmarks, each representing single robot arm manipulation and humanoid manipulation. For the real-world setting, we propose 4 pick-and-place tasks with the Franka Research 3 robot arm. We additionally evaluate DUST on the LIBERO (Liu et al., 2023) and CALVIN ABC-D (Mees et al., 2022) frameworks in Appendix A.1 (Results can be seen in Tables 9 and 10).

**RoboCasa kitchen.** RoboCasa is a single arm manipulation benchmark focusing on kitchen environment interaction. We utilize a suite of 24 tasks, including turning sink faucets, closing doors, and moving objects. The training dataset is drawn from the public dataset offered by RoboCasa (Nasiriany et al., 2024). We experiment over 100, 300, and 1000 training episodes per task as training data.

**GR-1 tabletop tasks.** GR-1 is a humanoid robot benchmark with a focus on dexterous tabletop manipulation of everyday objects. We utilize a total of 24 tasks, mostly comprised of pick-and-place tasks, with some tasks having additional articulated requirements, such as closing a drawer or microwave. The training dataset is taken from GR00T-N1.5 (NVIDIA et al., 2025). We experiment over 300 and 1000 training episodes per task as training data.

*Table 4.* **Evaluation of joint training.** Success rates (%) on Robo-Casa when joint training with or without GR-1 and EgoDex data mixture. For RoboCasa and GR-1, 300 demonstrations for each task is used, while we utilize a Fourier-hand action retargeted subset of EgoDex with 46k episodes. [†]: reproduced results.

| Method | Data Mixture | PnP | OP/CL | Other | Avg. |
|---|---|---|---|---|---|
| GR00T-N1.5 | ✗ | 27.2 | 66.0 | 46.6 | 45.0 |
| | ✓ | 30.2 | 69.6 | 46.6 | 46.9 |
| + FLARE[†] | ✗ | 38.0 | 76.7 | 56.2 | 55.3 |
| | ✓ | 41.1 | 76.9 | 59.2 | 57.6 |
| + DUST | ✗ | 42.3 | 80.7 | 58.1 | 58.5 |
| | ✓ | **50.2** | **82.1** | **65.2** | **64.4** |

*Table 5.* **Evaluation of pretraining setup.** Success rates (%) on RoboCasa benchmark comparing performance with or without video data pretraining. BridgeV2 data with only video input is used for the pretraining stage, while 100 RoboCasa demos per task are used for the finetuning stage. [†]: reproduced results.

| Method | Video Pretrain | PnP | OP/CL | Other | Avg. |
|---|---|---|---|---|---|
| GR00T-N1.5 | ✗ | 21.5 | 60.3 | 46.8 | 41.7 |
| + FLARE[†] | ✗ | 23.0 | 64.8 | 49.8 | 44.6 |
| | ✓ | 33.4 | 77.7 | 58.8 | 55.1 |
| **+ DUST** | ✗ | 29.5 | 76.0 | 51.0 | 50.1 |
| | ✓ | **42.3** | **80.7** | **58.1** | **58.5** |

**Real-world setup.** We conduct real-world experiments using a 7-DoF Franka Research 3 robotic arm, where both state and action spaces are parameterized by the arm's joint positions together with a binary gripper state. Evaluation is performed on a suite of four pick-and-place tasks, one insertion task, and two tool-using tasks in a tabletop setting. Detailed information on task configurations and evaluation is in Figure 7 and Appendix A.5. The training corpus consists of 60 expert demonstrations per task, gathered via teleoperation on the same Franka platform.

**Simulation results.** Tables 1 and 2 show that DUST consistently outperforms all baselines across both RoboCasa and GR-1 benchmarks, covering all task categories and demonstration scales. On RoboCasa with 100 demonstrations per task, DUST improves the average success rate by 18% over GR00T-N1.5 and 5% over FLARE, and this advantage remains as the number of demonstrations increases, confirming both data efficiency and scalability. On GR-1, a more challenging humanoid benchmark, DUST again surpasses all baselines at 300 and 1000 demonstrations, yielding improvements in both task categories.

**Real-world results.** Table 3 presents results on the Franka Research 3 robot across a suite of 7 diverse real-world tasks. DUST consistently outperforms baseline models, achieving the highest success rate on every task, ranging from standard pick-and-place behaviors to complex cord insertion to tool-mediated actions. On average, DUST achieves an improvement of 13% over the GR00T-N1.5 baseline and 10.4% over the FLARE-enhanced model. These wide-ranging improvements underscore DUST's robustness in physical environments and promise of deployment in practical settings.

### 5.2. Transfer learning

Acquiring task-specific robot demonstrations is prohibitively expensive, necessitating the use of heterogeneous robot mixtures or action-free video for scaling VLA models. Robot-centric datasets are growing in size (Khazatsky et al., 2024;

Bu et al., 2025a), and action-free video datasets are accessible via human recordings or web-scale crawling (Ye et al., 2025; Dass et al., 2023; Wang et al., 2025b). We demonstrate the efficacy of DUST across both paradigms: first, through joint training on robot-human mixtures, and second, via a pretraining-finetuning pipeline that adapts action-free representations to target embodiments.

**Joint training setup.** Robotic datasets are often heterogeneous, featuring diverse action spaces, degrees of freedom, and morphologies. This structural variance makes direct action regression difficult, as target output distributions often conflict across sources. To address this, we leverage DUST's world-modeling to bridge these discrepancies, aligning robotic and human data within a shared representational framework. In our setup, the baseline involves fine-tuning a model on the RoboCasa dataset using 300 demos per task. We further augment this training set with GR-1 data with 300 demos per task and the EgoDex human egocentric video dataset (Hoque et al., 2025). To integrate the human-centric data into the training pipeline, we employ hand pose estimation to extract MANO hand poses (Romero et al., 2017), which are subsequently retargeted to the Fourier hands action space.

Table 4 shows that joint training consistently improves performance across all evaluated methods, with DUST achieving the most significant gains. When augmented with the GR-1 and EgoDex data mixture, DUST's average success rate increases from 58.5% to 64.4%. These results demonstrate that DUST's world-modeling objective effectively leverages heterogeneous data sources to enhance policy robustness, outperforming both the GR00T-N1.5 baseline and the FLARE augmentation in its ability to translate cross-embodiment data into improved task success.

**Pretraining setup.** Leveraging large-scale video datasets allows models to acquire generalizable representations of object dynamics and scene evolution without relying on low-level action annotations. DUST's dual-stream architecture is naturally suited for this setting, as it enables pre-

*Table 6.* **Results of test-time scaling with asynchronous vision-action joint sampling.** Success rates (%) on RoboCasa and GR-1 with test-time scaling using asynchronous joint sampling. For scaling, we increase $N_o$, the number of diffusion steps for vision tokens.

| $N_o$ | RoboCasa 100 demos | | | | RoboCasa 1000 demos | | | | GR-1 1000 demos | | |
|---|---|---|---|---|---|---|---|---|---|---|---|
| | PnP | OP/CL | Other | Avg. | PnP | OP/CL | Other | Avg. | PnP | Art. | Avg. |
| 4 | 29.5 | 76.0 | 51.0 | 50.1 | 48.3 | 86.3 | 68.6 | 66.3 | 42.2 | 41.3 | 42.0 |
| 16 | 30.8 | 73.3 | 52.4 | 50.4 | 49.8 | 85.6 | 69.0 | 66.8 | 44.7 | 46.3 | 45.1 |
| 32 | 24.8 | 75.3 | 56.8 | 50.8 | 50.1 | 86.8 | 72.4 | 68.6 | 47.1 | 47.2 | **47.1** |
| 64 | 29.0 | 77.0 | 54.8 | **51.8** | 50.9 | 88.1 | 73.6 | **69.7** | 43.0 | 51.1 | 45.0 |

*Table 7.* **Ablation study.** Success rates (%) on RoboCasa benchmark with 100 demos/task ablating over (a) architecture and how noise is sampled during training, (b) depth of the MMDiT block stack, and (c) the loss weight $\lambda_{\text{WM}}$ for world-modeling loss.

(a) Arch. and Noise Sampling

| Arch. | Noise | PnP | OP/CL | Other | Avg. |
|---|---|---|---|---|---|
| DiT | Joint | 24.0 | 63.3 | 34.0 | 38.0 |
| DiT | Decoupled | 24.8 | 61.3 | 45.4 | 42.5 |
| MMDiT | Joint | 16.0 | 67.7 | 38.2 | 38.2 |
| MMDiT | Decoupled | 29.5 | 76.0 | 51.0 | 50.1 |

(b) MMDiT Depth

| Layers | Avg. |
|---|---|
| 6 | 47.4 |
| 10 | 48.3 |
| 12 | 50.1 |
| 14 | 49.3 |

(c) Loss Weight

| $\lambda_{\text{WM}}$ | Avg. |
|---|---|
| 0.2 | 34.3 |
| 0.5 | 48.9 |
| 1.0 | 50.1 |
| 2.0 | 49.6 |

training on action-free video to accumulate world-modeling knowledge prior to finetuning as a policy, thereby bridging the gap between inexpensive large-scale video data and costly teleoperated robot data. For this pipeline, during the pretraining stage, the model is trained exclusively on the video component of the BridgeV2 dataset (Walke et al., 2023), optimizing only the world-modeling term of the flow matching loss while randomly initializing the action tokens. After pretraining, we finetune the model on the RoboCasa dataset using 100 demonstrations per task. Table 5 shows that incorporating video pretraining yields substantial gains, with DUST achieving an average success rate of 58.5 compared to 50.1 without pretraining. These results highlight that large-scale passive video data can effectively transfer to downstream policy learning, improving data efficiency and generalization while reducing dependence on expensive robot demonstrations.

### 5.3. Test-time scaling for joint sampling

While our main experiments adopt the same number of diffusion steps for both actions and vision, this symmetry may not be optimal. The higher dimensionality and structural complexity of image embeddings typically requires more denoising iterations than the lower-dimensional and temporally smooth action tokens. To account for this, we introduce a test-time scaling strategy in which vision tokens are allocated additional diffusion steps while action token steps are fixed, thereby enabling finer-grained refinement of visual representations. We increase the number of vision denoising steps $N_o$ from its default value of 4 to 16, 32, and 64, while keeping the number of action token steps fixed at $N_A = 4$. Experiments are conducted using DUST checkpoints finetuned on RoboCasa with 100 and 1000 demonstrations per task, as well as GR-1 with 1000 demonstrations per task.

As shown in Table 6, increasing the number of vision denoising steps leads to mostly steady performance gains. On RoboCasa, we observe improvements of roughly 2–3% at 64 steps, while on GR-1 the best results occur at 32 steps with a 5% gain. These findings indicate that additional diffusion steps for vision tokens can enhance VLA performance by allowing more precise refinement of visual representations. However, the improvements come at the expense of higher inference time, highlighting a tunable trade-off. Further ablations on the role of modality decoupling in this process are provided in Section A.2. We note that performance gains are non-monotonic in some settings (*e.g.*, GR-1 at $N_o = 64$). This is a known trade-off in diffusion models using deterministic ODE solvers, where while increasing steps reduces truncation error, it simultaneously accumulates gradient approximation errors (Wang et al., 2025a) that compound without the corrective contraction of stochastic solvers. Importantly, test-time scaling is an *elective* mechanism for maximum precision. DUST with only 4 denoising steps already significantly outperforms baselines such as GR00T-N1.5 and FLARE while operating at ~40Hz (Table 4), well above the 10Hz threshold for dynamic closed-loop control. This flexibility allows us to reserve higher denoising steps for less time-sensitive tasks that demand maximum predictive accuracy.

### 5.4. Ablation study

**DUST components analysis.** We next conduct an ablation study to disentangle the contributions of DUST's two core design elements: the dual-stream MMDiT architecture and decoupled training algorithm. To this end, we evaluate 3 configurations: (1) a baseline DiT model trained with a uniform noise schedule, serving as a standard single-stream reference, (2) a DiT model with decoupled noising, where

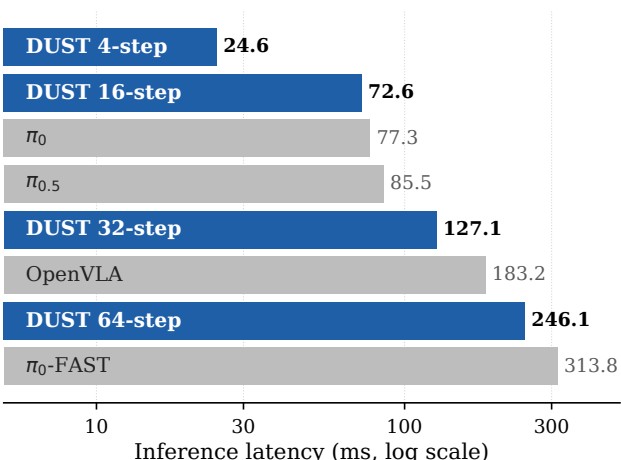

*Figure 4.* **Inference time.** Latency (ms) and control frequency on an RTX 5090 with TorchInductor compilation, compared with other VLA models.

AdaLN conditioning is applied independently to each modality, but the token streams still share a single feed-forward pathway, and (3) an MMDiT model with uniform noise levels. This design allows us to isolate the relative benefits of modality-specific noise schedules and of the dual-stream transformer structure itself. Results on RoboCasa with 100 demonstrations per task (Table 7.a) show that both components are indispensable. Removing the dual-stream structure results in a performance drop of 8%, while removing decoupled noise leads to a 12% reduction. These findings confirm that the two design choices contribute complementary gains, with MMDiT enabling structured cross-modal representation learning, while decoupled noising allows each modality to learn causal relationships.

**Loss weight hyperparameter $\lambda_{WM}$.** Table 7.c shows results for ablations on the loss weight. Larger $\lambda_{WM}$ values emphasize world-modeling, while smaller values emphasize action-modeling. Experiments on RoboCasa with 100 demos per task indicate that performance remains stable in the range $\lambda_{WM} \in [0.5, 2.0]$, but degrades when moving outside this interval. This suggests that effective learning requires weighting the two objectives relatively evenly.

**MMDiT layer count.** Table 7.b shows results for ablations on the number of MMDiT layers. Fixing the total layer count in $\pi_\theta$ to 16, we vary the number of MMDiT layers to adjust the trade-off between cross-modal transfer and per-modality specialization. Results show that the best outcome is obtained with 12 MMDiT layers, highlighting the benefit of leveraging cross-modal processing.

**Inference time.** Figure 4 reports latency benchmarked on an RTX 5090 GPU with TorchInductor compilation, providing a more accurate reflection of deployment conditions. DUST with 4 denoising steps achieves $\sim$40Hz, which is 3–10$\times$ faster than baseline VLA models such as $\pi_0$ ($\sim$13Hz), $\pi_0$-

*Table 8.* **Latent embedder ablation.** Success rates (%) on RoboCasa with 100 demos/task comparing world-modeling targets: semantic embeddings (SigLIP-2, DINOv2) vs. reconstructive latents (Flux.1 VAE).

| Latent Embedder | PnP | OP/CL | Other | Avg. |
|---|---|---|---|---|
| SigLIP-2 | 0.295 | 0.760 | 0.510 | 0.501 |
| DINOv2 | 0.281 | 0.770 | 0.552 | 0.516 |
| Flux.1 VAE | 0.285 | 0.700 | 0.530 | 0.491 |

FAST ($\sim$3Hz), and OpenVLA ($\sim$5.5Hz). This efficiency stems from the lightweight diffusion head operating over a frozen VLM backbone, whereas autoregressive methods iterate over a heavier VLM to generate each token. Even at 32 denoising steps for test-time scaling, DUST maintains $\sim$8Hz, which remains practical for closed-loop control.

**Latent embedder choice.** We chose SigLIP-2 embeddings as our world-modeling target because they capture the semantic structure of the environment. To validate this choice, we conduct a comparative study replacing SigLIP-2 with two alternatives: DINOv2 (Oquab et al., 2024) embeddings, which also provide semantic representations, and Flux.1 VAE (Black Forest Labs, 2024) latents, which are optimized for high-fidelity pixel-level reconstruction. As shown in Table 8, the Flux.1 VAE, despite its superior pixel-level reconstruction capability, does not yield better downstream performance than SigLIP-2 or DINOv2. This suggests that high-frequency visual details captured by VAEs may be irrelevant, or even distracting, for policy learning.

## 6. Conclusion

In this work, we presented **du**al-**st**ream diffusion (DUST), a world-model augmented VLA framework that resolves modality conflicts by maintaining separate, attention-linked token streams for actions and observations. By utilizing a decoupled training algorithm with independent noising, DUST captures causal dependencies to achieve gains in simulated benchmarks and real-world tasks. Our asynchronous joint sampling strategy enables scalable test-time refinement, while action-free video pretraining and heterogeneous joint training show successful transfer learning.

## Acknowledgements

This work was partly supported by Institute for Information & communications Technology Planning & Evaluation (IITP) grant funded by the Korea government (MSIT) (RS-2019-II190075, Artificial Intelligence Graduate School Support Program(KAIST); RS-2025-02653113, High-Performance Research AI Computing Infrastructure Support at the 2 PFLOPS Scale; RS-2024-00509279, Global AI Frontier Lab) and RLWRLD Inc. We also thank Jimin Lee and Suheyok Jang for their invaluable support in designing and conducting real-world experiments.

## Impact Statement

This paper presents work whose goal is to advance the field of Machine Learning by improving the physical grounding and world-modeling capabilities of Vision-Language-Action (VLA) models. There are many potential societal consequences of our work, none which we feel must be specifically highlighted here.

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

*Table 9.* **Evaluation on LIBERO benchmark** Success rates (%) on LIBERO benchmark across Long, Goal, Object, and Spatial task categories. [†]: reproduced results.

| Method | LONG | GOAL | OBJECT | SPATIAL | Avg. |
|---|---|---|---|---|---|
| DreamVLA | 89.5 | 89.5 | 94.0 | 97.5 | 92.6 |
| WorldVLA | 60.0 | 83.4 | 96.2 | 87.6 | 81.8 |
| FlowVLA | 72.6 | 91.6 | 95.0 | 93.2 | 88.1 |
| CoT-VLA | 69.0 | 87.6 | 91.6 | 87.5 | 81.1 |
| UD-VLA | 89.6 | 91.2 | 95.7 | 94.1 | 92.7 |
| $\pi_0$ | 85.2 | 92.6 | 97.8 | 96.0 | 92.9 |
| $\pi_0$-FAST | 78.8 | 88.2 | 96.0 | 93.0 | 89.0 |
| GR00T-N1.5 | 83.0 | 95.6 | 99.6 | 92.8 | 92.8 |
| + FLARE[†] | 92.2 | 95.6 | **100.0** | **96.8** | **96.2** |
| + DUST | **92.6** | **96.0** | 99.8 | 96.2 | **96.2** |

*Table 10.* **Evaluation on CALVIN ABC-D benchmark** Success rates (%) of tasks completed in a row and average length of successful trajectories on CALVIN ABC-D benchmark. [†]: reproduced results.

| Method | Task Completed in a Row | | | | | Avg. |
|---|---|---|---|---|---|---|
| | 1 | 2 | 3 | 4 | 5 | |
| UP-VLA | - | - | - | - | - | 2.74 |
| Seer | 93.0 | 82.4 | 72.3 | 62.6 | 53.3 | 3.64 |
| MDT | 63.1 | 42.9 | 24.7 | 15.1 | 9.1 | 1.55 |
| GR00T-N1.5 | 55.8 | 25.9 | 10.7 | 4.3 | 1.3 | 0.98 |
| + FLARE[†] | **96.0** | 86.1 | 74.8 | 63.8 | 54.4 | 3.75 |
| + DUST | 93.8 | **86.5** | **78.2** | **70.5** | **62.3** | **3.91** |

# A. Appendix

## A.1. Additional simulation environments

**LIBERO.** We report results on the LIBERO benchmark (Liu et al., 2023) in Table 9 with comparison to DreamVLA (Zhang et al., 2025b), WorldVLA (Cen et al., 2025), FlowVLA (Zhong et al., 2025), CoT-VLA (Zhao et al., 2025), UD-VLA (Chen et al., 2025b), $\pi_0$ (Black et al., 2025), $\pi_0$-FAST (Pertsch et al., 2025), GR00T-N1.5 (NVIDIA et al., 2025), and FLARE (Zheng et al., 2025). The results for DreamVLA, WorldVLA, FlowVLA, CoT-VLA, and UD-VLA are taken from the values reported in the corresponding papers. For $\pi_0$ and $\pi_0$-FAST, we follow the training procedure similar to that in our main experiments, where we only initialize the Paligemma (Beyer et al., 2024) VLM backbone instead of the robot-pretrained checkpoint. All experiments that were done by us were configured with 32 global batch size, with 60k total training iterations. For the $\pi_0$ and $\pi_0$-FAST experiments, we utilize 4 A100 GPUs, and for the GR00T-N1.5, FLARE, and DUST experiments we utilize 2 A100 GPUs.

**CALVIN baselines.** We report results on the CALVIN ABC-D benchmark (Mees et al., 2022) in Table 10 in comparison with UP-VLA (Zhang et al., 2025a), Seer (Tian et al., 2025), MDT (Reuss et al., 2024), GR00T-N1.5 (NVIDIA et al., 2025), and FLARE (Zheng et al., 2025). Results for UP-VLA and Seer are taken from their respective papers, where we utilize the ablation study results that exclude large-scale pretraining in order to ensure fair comparison with our settings. The MDT results are taken from the Video Prediction Policy (Hu et al., 2025) paper. For the GR00T-N1.5, FLARE, and DUST experiments, we trained on CALVIN-ABC with 32 global batch size over 200k total training iterations. We utilize 2 A100 GPUs during training.

## A.2. Test-time scaling of naive joint sampling

In Section 5.3, we explored test-time scaling DUST by increasing $N_o$, the number of vision token diffusion steps, while keeping $N_A$, the action diffusion step count, fixed at 4. While we have seen great performance gains through the asynchronous joint sampling, it is natural to ask whether simply increasing diffusion steps for both modalities could be enough.

In Table 11, we present results from an ablation study, where both $N_A = N_o$ are increased together, instead of fixing $N_A$

*Table 11.* **Results of test-time scaling with synchronous joint sampling.** Success rates (%) on RoboCasa and GR-1 with our test-time scaling approach using synchronous joint sampling. For scaling, we increase both $N_o$, $N_A$, the number of diffusion steps for vision tokens and action tokens, respectively.

| $N_o$ | RoboCasa 100 demos | | | | RoboCasa 1,000 demos | | | | GR-1 1000 demos | | |
|---|---|---|---|---|---|---|---|---|---|---|---|
| | PnP | OP/CL | Other | Avg. | PnP | OP/CL | Other | Avg. | PnP | Art. | Avg. |
| 4 | 0.295 | 0.760 | 0.510 | **0.501** | 0.483 | 0.863 | 0.686 | **0.663** | 0.422 | 0.413 | **0.420** |
| 16 | 0.197 | 0.685 | 0.450 | 0.425 | 0.472 | 0.854 | 0.621 | 0.630 | 0.402 | 0.443 | 0.416 |
| 32 | 0.210 | 0.710 | 0.424 | 0.424 | 0.450 | 0.807 | 0.630 | 0.614 | 0.406 | 0.438 | 0.406 |
| 64 | 0.181 | 0.654 | 0.416 | 0.397 | 0.460 | 0.817 | 0.601 | 0.608 | 0.399 | 0.405 | 0.401 |

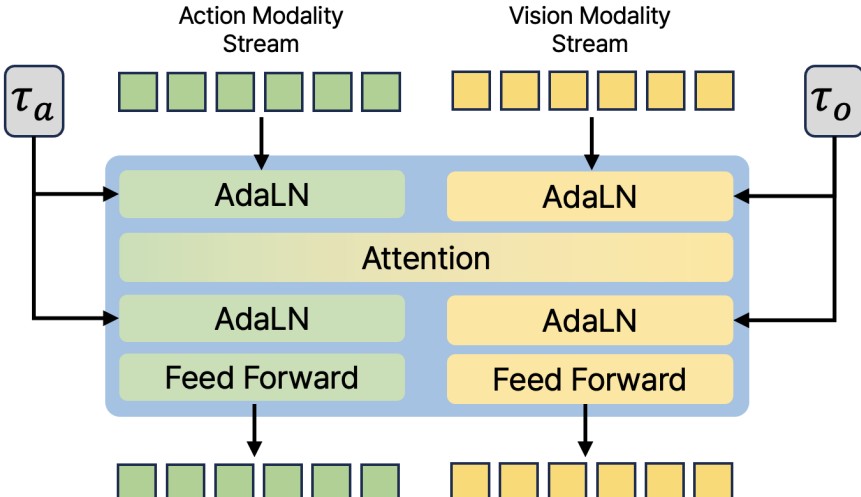

*Figure 5.* **Modified MMDiT.** DUST's MMDiT blocks are implemented with separate timestep embeddings being used as conditions for each modality.

and increasing $N_o$. We can see that without the decoupling of number of steps between modalities, simply increasing diffusion steps actually leads to deterioration in performance. This lends credibility to our initial hypothesis of only vision tokens needing more diffusion steps, and shows that the asynchronous component of our test-time scaling method is crucial to its success.

### A.3. Implementation and Training Details

**Additional implementation details.** We base our architecture on the GR00T-N1.5 (NVIDIA et al., 2025) codebase, from which we get the pretrained Eagle-2 VLM model. For vision tokens, they pass through an encoder made up of a 3-layer MLP with 2D sinusoidal positional encoding with SiLU activation. The vision decoder is a 2-layer MLP with ReLU activation. Action tokens utilize the linear encoder-decoder pair given in the original code-base, alongside 1D sinusoidal positional encoding.

The MMDiT blocks used in our model are a slight modification of the original in that the AdaLN layers for each modality stream take the conditioning timestep embeddings from independent sources instead of utilizing a global timestep embedding. We show this in more detail in Figure 5.

**Baselines.** The GR00T-N1.5 baseline is trained on the original released code, while the FLARE baseline does not release official code or checkpoints. Hence, for FLARE, we do not utilize the Q-Former architecture of the original paper, but re-implement the FLARE loss to utilize the same world modeling target as ours, which is the SIGLIP-2 embeddings from the model VLM. This allows fair comparison between dual-stream diffusion world modeling of DUST and the implicit world modeling of FLARE. For the alignment module of FLARE we use a small MLP, with similar architecture to that of REPA (Yu et al., 2025), which inspired FLARE. For PAD (Guo et al., 2024) and VPP (Hu et al., 2025), we utilize the base training and evaluation configuration settings given in the released code bases, using the default prediction horizon, batch

size, epoch count etc.

**Batch size and iteration count.**   We vary batch size and training time per dataset.

- For the RoboCasa (Nasiriany et al., 2024) dataset, we train using global batch size 32, with 2 A100 GPUs. For each training dataset scale, the time until convergence varies, with 100 demos requiring 60k steps, 300 demos requiring 420k steps, and 1000 demos requiring 600k steps. The long convergence time is mostly due to the small global batch size.

- For the GR-1 (NVIDIA et al., 2025) dataset, we train using global batch size of 960, with 8 H200 GPUs over 60k steps. We noted training on GR-1 was very sensitive to batch size and required large scale training for meaningful training results.

- For the real-world dataset, we train using global batch size of 32, with 2 A100 GPUs over 60k steps. For $\pi_0$ and $\pi_0$-FAST, we maintain the global batch size of 32, but utilize 4 A100 GPUs due to higher memory usage.

- For the joint training setup, we train with a mixture of RoboCasa 300 demo dataset, GR-1 300 demo dataset, and 46k trajectories from EgoDex. We train with a global batch size of 512, with 8 A100 GPUs for 60k steps.

- For the action-free video pretraining setup, we first train with BridgeV2 (Walke et al., 2023) video data using global batch size of 32, with 2 A100 GPUs for 120k steps. Then, we finetune using the RoboCasa 100 demo dataset with the same GPU setup for 60k steps.

**Common training details.**   Excluding batch size and iteration count, all experiments are done with the same training hyperparameters. We optimize with AdamW (Loshchilov & Hutter, 2019) using a base learning rate of 1e-4, with $\beta_1 = 0.95$, $\beta_2 = 0.999$, and $\epsilon = $ 1e-8. Weight decay of 1e-5 is applied with the exception of bias and LayerNorm weights. The learning rate follows a cosine decay schedule with a 5% warmup period.

### A.4. Simulation Benchmarks

**RoboCasa kitchen.** RoboCasa is a single arm manipulation benchmark with a focus on kitchen environment interaction tasks. We utilize a suite of 24 tasks that span a wide range of common household manipulations, including turning sink faucets, closing drawer doors, and moving objects. Tasks are categorized into 8 pick-and-place tasks, 6 contraption open/close tasks, and 10 other miscellaneous tasks. Training data is drawn from the publicly available dataset from RoboCasa which was generated with MimicGen (Mandlekar et al., 2023) within the MuJoCo simulation environment (Todorov et al., 2012), with a Franka Emika Panda robot arm serving as the manipulator. Image observations include 3 viewpoints from the left, right, and wrist. The robot state/action space is parameterized with 7 degrees of freedom (DoF), consisting of end-effector position and rotation together with a binary gripper pose. We experiment over 100, 300, and 1000 training episodes per task, testing data efficiency and scaling properties.

**GR-1 tabletop tasks.** GR-1 is a humanoid robot benchmark with a focus on dexterous tabletop manipulation of everyday objects. We utilize a total of 24 tasks consisting of 16 pick-and-place tasks, and 8 articulated tasks, the latter adding the requirement of closing containers such as microwaves and cabinets after pick-and-place. Training data utilizes data from GR00T-N1.5 (NVIDIA et al., 2025), where the dataset was generated with DexMimicGen (Jiang et al., 2025) in the MuJoCo simulation environment (Todorov et al., 2012). The simulated robot is a GR-1 humanoid robot with Fourier dexterous hands, enabling fine-grained grasping and manipulation. Image observations are taken from a single egocentric view from the robot's head. The state/action space consists of 29 DoF in total, 17 DoF corresponding to the GR-1 robot's arms and waist, and 6 DoF for each of the Fourier hands. We experiment over 300 and 1000 training episodes per task.

### A.5. Real-World Experiment Details

We utilize a total of 7 tasks: 4 pick-and-place tasks, 1 insertion task, and 2 tool using tasks.

**Pick-and-place tasks.**   Our pick-and-place tasks consist of 4 tasks, which have the following task instruction templates:

- Pick up the {*Object*} on the brown box and place it in the golden bowl.

- Pick up the {*Object*} on the brown box and place it on the white plate.

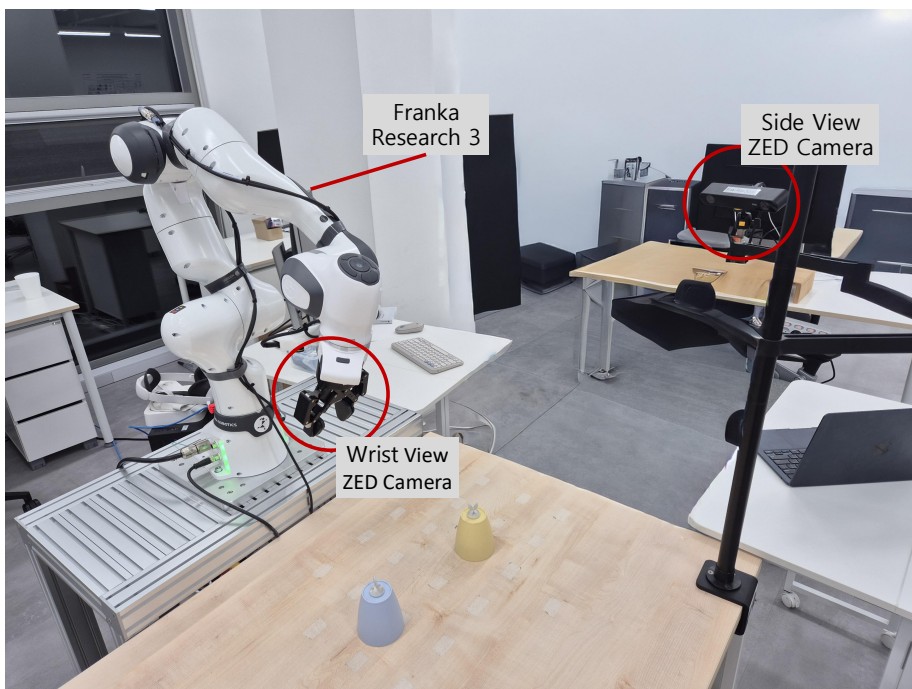

*Figure 6.* **Real-world experimental setting.** We utilize the Franka Research 3 robot with two ZED cameras, one on the wrist and one to the side.

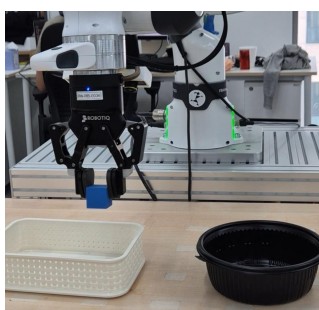 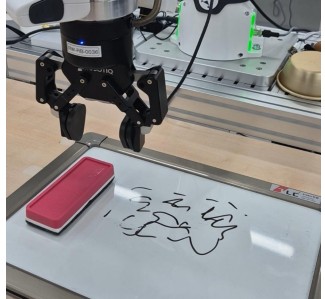 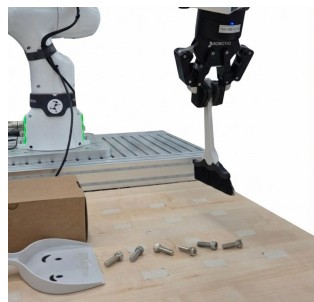 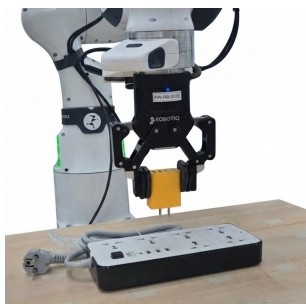

*PnP Blue Cube from white basket to black bowl*    *Use the red eraser to clean the board.*    *Pick up the brush and clean the bolts into the dustpan.*    *Pick up the yellow charger and plug it into the socket.*

*Figure 7.* **Real-world task instructions.** For the real-world experiments, we utilize the Franka Research 3 robot. The task suite is composed of 4 pick-and-place (PnP) tasks, 2 tool using tasks, and 1 insertion task. The PnP tasks are categorized by their distinct source-target pairs (box, bowl, plate, etc.) and each contains 4 different objects (cup, doll, cube, sponge). The tool-using tasks require more complex motions such as sweeping or swiping, while the insertion task requires precise manipulation under occlusion of the wrist camera by a large object.

- Pick up the {*Object*} in the white basket and place it in the black bowl.

- Pick up the {*Object*} on the white plate and place it in the white basket.

Each task contains the four object categories - Teddy Bear, Blue Cube, Blue Cup, and Sponge. During evaluation each object-task configuration gets 6 evaluations, meaning 24 trials per task. We predetermine a set of varied configurations of where to place the source-target locations, on where the source location the object is placed, and the direction it is facing. This allows for more fair comparison in real-world experiments that typically have high stochasticity. When an object has been partially placed in the target destination but the center of gravity is outside of said target, we denote that as a half success and count it as 0.5 successes. We note there were very few cases of this happening.

**Insertion task.** We have one precise insertion task, which has the following task:

- Pick up the yellow charger and plug it into the socket.

We evaluate over 4 different configurations, with 6 evaluations, totaling 24 trials. There is no partial score for the insertion task.

**Tool using tasks.**  We have two tool-using tasks, which have the following instructions:

- Use the red eraser to clean the board.

- Pick up the brush and clean the bolts into the dustpan.

For the eraser task, we utilize two different configurations with 12 evaluations each. In the case where we erase over 50% of the marker prints, we give half points, and give full points when we erase over 90% of the prints. For the brush task, for each of the 24 evaluations, the aim is to clean up 6 bolts into a dustpan. We give $1/6$-th of a point for each bolt that ends up inside of the dustpan.

## A.6. Example GR-1 Rollouts

We showcase example rollouts of DUST trained on GR-1 with 1000 demos per task.

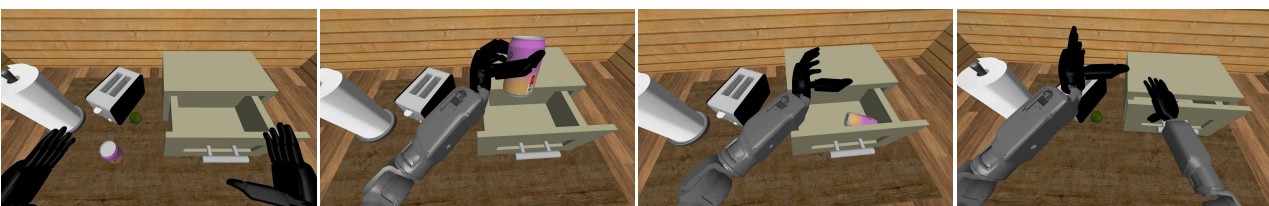

*(a)* **(GR-1)** *Pick up the can, place it into the drawer and close the drawer.*

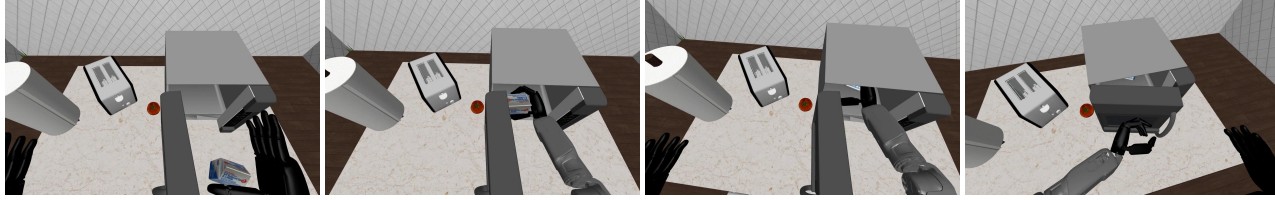

*(b)* **(GR-1)** *Pick up the milk, place it into the microwave and close the microwave*

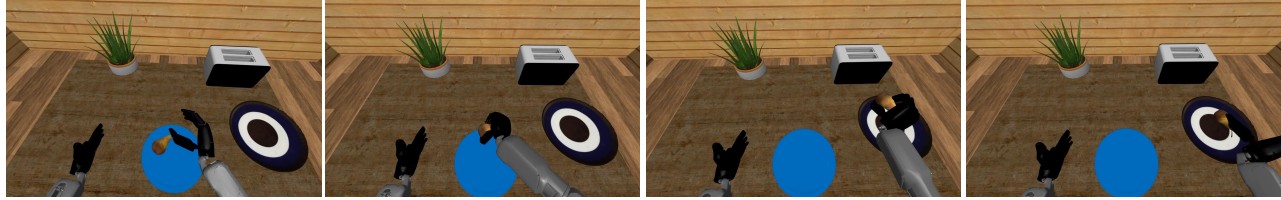

*(c)* **(GR-1)** *Pick the pear from the plate and place it in the plate*

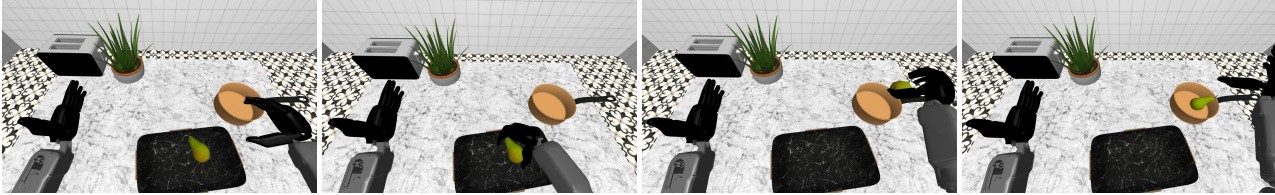

*(d)* **(GR-1)** *Pick the pear from the tray and place it in the pot*

## A.7. Example RoboCasa Rollouts

We showcase example rollouts of DUST trained on RoboCasa with 1000 demos per task.

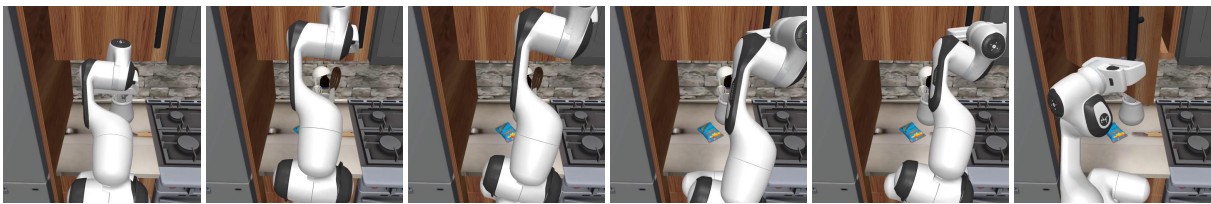

*(a)* **(RoboCasa)** *Open the cabinet door*

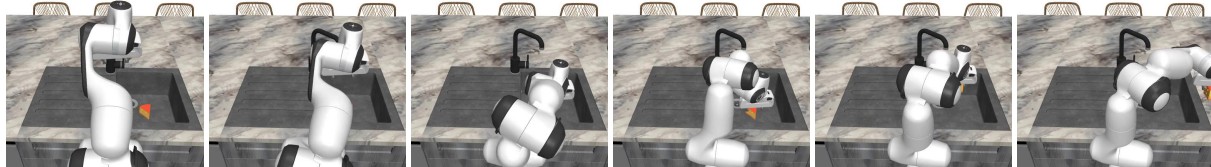

*(b)* **(RoboCasa)** *Pick the cheese from the sink and place it on the plate located on the counter*

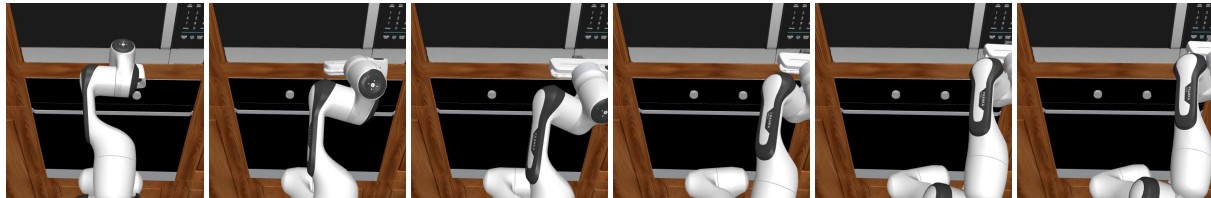

*(c)* **(RoboCasa)** *Turn on the microwave*

## A.8. Example Real-world Rollouts

We showcase example rollouts of DUST trained on our real-world Franka Research 3 dataset with 60 demos per task.

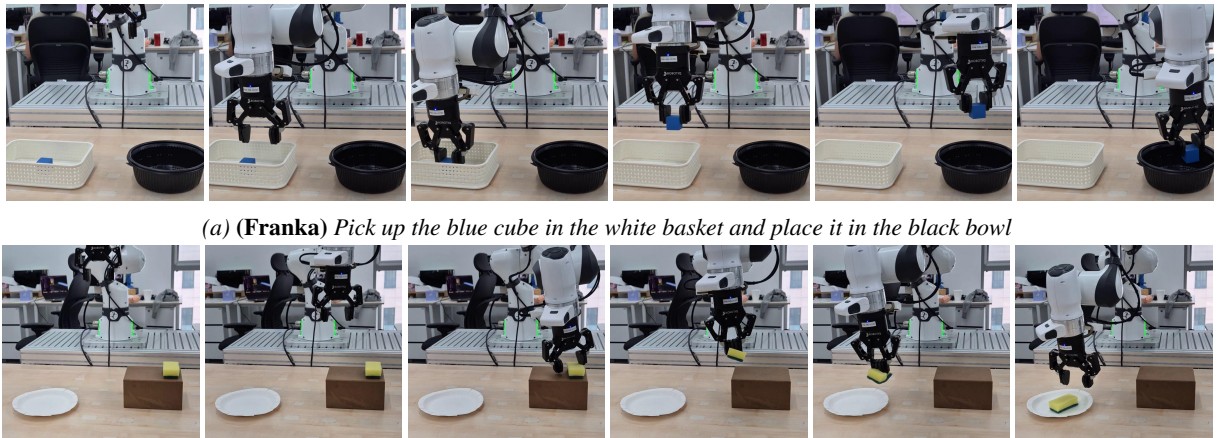

*(a)* **(Franka)** *Pick up the blue cube in the white basket and place it in the black bowl*

*(b)* **(Franka)** *Pick up the sponge on the brown box and place it on the white plate*

### A.9. Action-vision causal dependency verification

To verify that DUST has learned a meaningful joint distribution between modalities, we evaluate action generation accuracy when conditioning on ground-truth future states with varying levels of added noise. This experiment is conducted in the RoboCasa setting.

*Table 12.* **Action generation accuracy under noisy future states.** MSE of predicted actions when conditioning on ground-truth future observations corrupted with different noise levels.

| Noise Level | Action Generation Accuracy (MSE) |
| --- | --- |
| 0 (Clean) | **0.0318** |
| 0.05 | 0.0436 |
| 0.1 | 0.0507 |
| 0.2 | 0.0642 |
| 0.5 | 0.0816 |
| 1.0 | 0.0865 |

The results in Table 12 show a clear monotonic degradation in action accuracy as the noise level of the future state increases, confirming that the model effectively leverages the learned dependency between modalities to guide action generation. This demonstrates that DUST has captured meaningful cross-modal causal relationships, rather than relying on a surface-level alignment.

### A.10. Zero vision denoising steps

We investigate the extreme case of running zero vision denoising steps during inference, effectively disabling explicit visual prediction at test time.

*Table 13.* **Zero vision denoising steps.** Success rates (%) on RoboCasa with 100 demos/task when vision denoising is disabled during inference.

| Vision Denoising Steps | PnP | OP/CL | Other | Avg. |
| --- | --- | --- | --- | --- |
| 4 (Default) | 29.5 | 76.0 | 51.0 | **50.1** |
| 0 | 19.7 | 78.7 | 51.9 | 47.9 |

As shown in Table 13, the average performance degrades slightly compared to the 4-step default. However, the zero-step performance (47.9%) still exceeds both the GR00T-N1.5 baseline (41.7%) and FLARE (44.6%). This suggests that DUST's world-modeling training objective guides the internal representations of the MMDiT to produce more robust actions even without explicit visual reconstruction at inference time, enabling an ultra-low-latency mode when needed.

## A.11. Failure case analysis

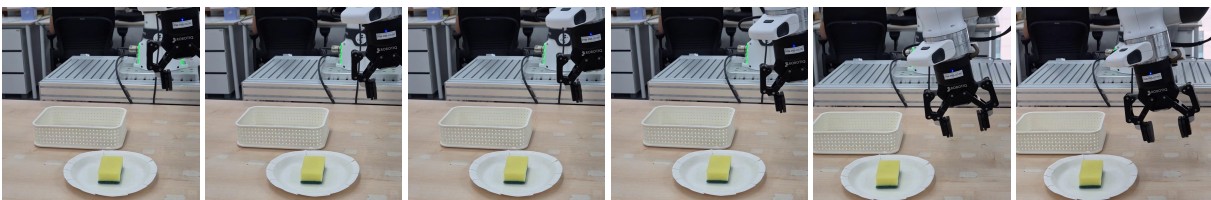

The above rollout illustrates a failure in precise grasping with our DUST model in Task 4 of the Pick-and-Place real-robot tasks. The robot arm approaches the sponge but fails to successfully grasp it or align its gripper correctly. Our real-world setup uses a Franka Research 3 arm with two ZED cameras: one on the wrist and one side view. In this particular rollout, the side view ZED camera is positioned such that it is partially obstructed by the robot arm itself, while the wrist view is positioned in a way in which none of the target objects are in view.

DUST is a world-model augmented VLA that explicitly predicts future visual states to guide its action generation. This explicit prediction allows DUST to anticipate where the gripper will land and adjust for better alignment, leading to higher success rates than models without world modeling (like GR00T-N1.5). However, this reliance on visual prediction makes DUST more prone to failure when the visual input is poor or occluded. If the current observation ($o_t^v$) or the prediction of the future observation ($\tilde{o}_{t+k}$) is compromised due to occlusion, DUST's key strength of anticipatory control is directly undermined, causing it to fail, as seen in the rollout. The reason for the higher failure rate in Task 4 is due to the specific source-target configuration used for this task which involves more self-occlusion by the arm or gripper.

Future directions that could help mitigate this problem include adding more camera views or stronger priors on proprioceptive robot state. Another direction could be integrating history (multiple past observations/actions) to provide temporal context for world modeling, making the prediction more robust against momentary visual noise or occlusion. The current DUST formulation primarily focuses on prediction from the current state.

