# OpenReview forum: "Dual-Stream Diffusion for World-Model Augmented Vision-Language-Action Model"
_ICML.cc/2026/Conference — ICML 2026 regular_

### Official Review · Reviewer_EXJ2 · 2026-03-02

**Soundness:** 3
**Presentation:** 4
**Significance:** 3
**Originality:** 2
**Overall Recommendation:** 5
**Confidence:** 4

**Summary:**

The authors propose a dual-stream diffusion architecture for joint action and visual state prediction for robotic applications.  They build separate diffusion streams for each modality with joint-training. Modality specific noising during training allows multiple train objectives (e.g. resembling inverse and forward dynamics). During inference, their setup allows modality specific denoising step counts, allowing flexible test-time scaling. Thorough experiments and ablations on simulation and real-world environments establish the strong performance of their method.

**Compliance With Llm Reviewing Policy:**

Affirmed.

**Key Questions For Authors:**

1. What happens if zero vision denoising steps are run? Can the model run without generating the visual future state? This would be like the opposite of test-time scaling (or rather scaling to less compute for very fast inference).

**Strengths And Weaknesses:**

**Strengths**
1. Clear problem formulation and motivation
2. Simple architectural modification leading to improvements
3. Extensive experimentation to establish strong performance
4. Thorough ablations of each design choice and claim
5. Minimal inference time increase over baseline

**Weaknesses**
1. Novelty maybe incremental over PAD, FLARE
2. Non-VAE embedding based world model target might miss details of full pixel reconstruction
3. Unclear why only 4 action denoising steps are used (vs 20+ in more diffusion policy works)

**Minor Related Work Suggestions**
1. "VideoJAM: Joint Appearance-Motion Representations for Enhanced Motion Generation in Video Models" - consider discussing their dual denoising strategy for RGB vs Flow modalities; this resembles the DUST modality specific denoising.
2. "Future Optical Flow Prediction Improves Robot Control & Video Generation" - consider discussing the sequential visual then action denoising approach in this work. This is an alternative (dual diffusion setup) where the visual denoising can still undergo test-time scaling, but in a different way.

---

> ### Author Rebuttal · Authors · 2026-03-31
>
> Dear reviewer EXJ2,
>
> Thank you for your valuable comments and suggestions in reviewing our work. We address each of your questions and concerns individually as follows.
>
> ---
>
> ## [W1] Novelty over PAD, FLARE
>
> We clarify that DUST introduces fundamental architectural and algorithmic shifts that resolves the modality gap problem inherent in these previous works.
>
> PAD utilizes a unified architecture that concatenates action and vision tokens into a single stream. This forces disparate modalities into the same latent space, often leading to a mismatch between low-dimensional action trajectories and high-dimensional, complex visual data. DUST maintains distinct modality streams using an MMDiT architecture, allowing each modality to preserve its specific statistical structure while sharing knowledge through cross-modal attention.
>
> FLARE performs implicit world-modeling by aligning mid-level features. In contrast, DUST performs explicit joint world-modeling, utilizing a decoupled flow matching loss and independent noise schedules. This design forces the model to learn true bidirectional causal dependencies, allowing the predicting of the consequences of actions and the actions required to reach specific states, which implicit alignment alone does not guarantee.
>
> ---
>
> ## [W2] Pixel-level detail omitted
>
>
> We chose SigLIP-2 embeddings as our target because they capture the semantic structure of the environment. To test your concerns, we conducted a comparative study replacing our SigLIP-2 embeddings with Flux.1 VAE latents and DINOv2 embeddings, which are specifically optimized for high-fidelity reconstruction and semantic embeddings, respectively.
>
> \begin{array}{l|ccc|c}
> \hline
> \text{Latent Embedder} & \text{PnP} & \text{OP/CL} & \text{Other} & \text{Avg.} \newline
> \hline
> \text{SigLIP-2 (Semantic)} & 0.295 & 0.760 & 0.510 & 0.501 \newline
> \text{DINOv2 (Semantic)} & 0.281 & 0.770 & 0.552 & {\bf 0.516} \newline
> \text{Flux.1 VAE (Reconstructive)} & 0.285 & 0.700 & 0.530 & 0.491 \newline
> \hline
> \end{array}
>
> As shown in the table, the Flux.1 VAE, despite its superior pixel-level reconstruction capability, does not yield better downstream performance than SigLIP-2 or DINOv2. This suggests that high-frequency visual details captured by VAEs may be irrelevant, or even distracting, for policy learning.
>
> ---
>
> ## [W3] Why use 4 denoising steps?
>
> In recent diffusion-based VLA models, optimal denoising steps are not universal and they are typically found empirically for each specific model architecture. For instance, while some policies use higher counts, GR00T-N1.5 utilizes 4 steps, and the PI_0 model utilizes 10 steps.
>
> We adopted the 4-step setting to maintain consistency with our primary baseline, GR00T-N1.5. Using more steps for actions often yields diminishing returns or even performance degradation, as shown in our synchronous scaling ablation in Table 11 in the appendix.
>
> ---
>
> ## [Q1] What if 0 vision denoising steps?
>
> We thank the reviewer for this insightful suggestion and provide results for zero vision denoising steps below:
>
> \begin{array}{l|ccc|c}
> \hline
> \text{Vision Denoising Steps} & \text{PnP} & \text{OP/CL} & \text{Other} & \text{Avg.} \newline
> \hline
> \text{4 (Default)} & 0.295 & 0.760 & 0.510 & {\bf 0.501} \newline
> \text{0} & 0.197 & 0.787 & 0.519 & 0.479 \newline
> \hline
> \end{array}
>
> As expected, the average performance degrades slightly compared to the 4-step denoising baseline. However we note that this performance is still higher compared to the baseline GR00T-N1.5 or the FLARE variant. This suggests that DUST’s world-modeling training guides the internal representations of the MMDiT to generate more robust actions even without explicit visual reconstruction.
>
> ---
>
> ## Related work suggestions
>
> We thank the reviewer for the suggestions and will incorporate the papers to the related works section in the final version of the paper.

---

> > ### Author Rebuttal · Reviewer_EXJ2 · 2026-04-04
> >
> > Authors address all concerns. Retaining my vote for accept.
> >
> > Congrats to the authors on this great project!

---

### Official Review · Reviewer_R5Wh · 2026-03-12

**Soundness:** 2
**Presentation:** 3
**Significance:** 2
**Originality:** 2
**Overall Recommendation:** 4
**Confidence:** 4

**Summary:**

The paper proposes Dual-Stream Diffusion (DUST), a world-model augmented Vision-Language-Action (VLA) framework. To address the modality gap between high-dimensional visual observations and low-dimensional actions, DUST utilizes a Multimodal Diffusion Transformer (MMDiT) that processes action and vision tokens in separate streams connected via shared cross-attention layers. The authors introduce a decoupled training algorithm with independent noise schedules for each modality. Additionally, an asynchronous joint sampling strategy is proposed to allow test-time scaling by updating vision tokens more frequently than action tokens during inference. The method is evaluated on RoboCasa, GR-1, and a real-world Franka arm, showing empirical gains over baseline VLAs.

**Compliance With Llm Reviewing Policy:**

Affirmed.

**Final Justification:**

My concern has been resolved. I'm raising my score to 4.

**Key Questions For Authors:**

1. The core architectural contribution, separating modalities into two streams using MMDiT, is a well-established technique in other generative domains (e.g., text-to-image). Adapting this to VLAs by substituting text/image with action/vision is a straightforward engineering choice.

2. The proposed test-time scaling method, which asynchronously updates the vision and action tokens at different frequencies, appears to be an intuition-driven heuristic lacking rigorous theoretical justification. Notably, Table 6 demonstrates non-monotonic behavior as the computational budget increases. For example, on the RoboCasa 100 demos task, the average success rate drops from 52.4% at $N_o=16$ to 50.8% at $N_o=32$, and similarly on GR-1 from 47.1% at $N_o=32$ to 45.0% at $N_o=64$. This performance degradation under increased computation strongly suggests that the model has not fundamentally learned the true joint distribution between the two modalities. Instead, it seems to rely on a fragile, surface-level alignment that breaks down when test-time distributional shifts (caused by mismatched sampling steps) are pushed too far.

3. The framework forces the generation of high-dimensional future vision tokens alongside actions during inference due to the cross-attention dependency. While the authors claim real-time capability with a 0.104s latency , this strictly applies to the baseline 4-step setting ($N_o=4$). To achieve the touted performance gains via test-time scaling, the vision denoising steps ($N_o$) must increase to 32 or 64. However, the paper omits the actual latency metrics for these scaled settings. Because the action generation cannot bypass the vision stream, scaling $N_o$ to 64 would drastically multiply the inference time far beyond the ~0.1s threshold required for typical 10Hz closed-loop control. Consequently, the performance gains from test-time scaling are practically inaccessible, as they come at the cost of rendering the model unusable for real-world robotic deployment.

**Limitations:**

Yes. The authors have adequately discussed the limitations of their work. Specifically, in Appendix A.9, they provide a candid failure case analysis demonstrating that DUST's core mechanism.

**Strengths And Weaknesses:**

1. The paper features robust experimental validation across two distinct simulation benchmarks (RoboCasa and GR-1) and real-world robot deployment . The inclusion of transfer learning experiments (pretraining on action-free video and joint-training with human data) is a strong practical addition .

2. The paper is well-structured, and the visual comparisons of different diffusion architectures (Figure 1 and 2) effectively communicate the method's design .

---

> ### Author Rebuttal · Authors · 2026-03-31
>
> Dear reviewer R5Wh,
>
> Thank you for your valuable comments and suggestions in reviewing our work. We address each of your questions and concerns individually as follows.
>
> ---
>
> ## [W1] MMDiT novelty
>
> We emphasize that our primary contribution is not the naive adoption of a single component, but rather the integrated DUST framework designed to resolve the inherent modality conflicts in joint world-modeling. As demonstrated in our ablation studies, a naive adaptation of the MMDiT architecture shows low performance. Our success is fundamentally tied to the synergy between the dual-stream transformer, independent noise sampling, and the decoupled flow matching loss, which together force the model to learn bidirectional causal relationships.
>
> While the MMDiT architecture is established in T2I domains, DUST represents the first successful utilization of this structure for world-model augmented action generation. We believe that adapting these concepts to address the specific challenges of high-dimensional vision data and low-dimensional action trajectories is a non-trivial advancement.
>
> ---
>
> ## [W2] Non-monotonic performance gains with test-time scaling
>
> We first would like to clarify a misunderstanding. While we acknowledge that the GR-1 setting has a performance drop, we clarify that for the RoboCasa 100 demos task, performance increases from 50.4 to 50.8 when going from N=16 to N=32, as can be seen in Table 6.
>
> Overall, the phenomenon of non-monotonic performance gains is not an indication of fragile alignment, but rather a known trade-off in diffusion modeling using deterministic solvers. Our inference utilizes Euler integration over the learned velocity field, where increasing steps reduces truncation error, while simultaneously accumulating small approximation errors, often termed gradient error [1], present in the network's output. Because deterministic ODE solvers lack the corrective contraction found in SDE solvers, these errors compound at higher step counts.
>
> Furthermore, we clarify that distributional shift due to mismatched sampling steps is explicitly mitigated by our decoupled training algorithm. By independently sampling timesteps during training, we force the model to handle diverse, mismatched noise configurations, ensuring the model learns the underlying causal structure rather than a surface-level mapping.
>
> To demonstrate that DUST has captured the joint distribution, we performed an experiment comparing action generation accuracy when conditioned on static ground-truth future states with varying levels of noise. This experiment was conducted in the RoboCasa setting.
>
> \begin{array}{l|c}
> \hline
> \text{Noise Level} & \text{Action Generation Accuracy (MSE)} \newline
> \hline
> \text{0 (Clean)} & {\bf 0.0318} \newline
> \text{0.05} & 0.0436 \newline
> \text{0.1} & 0.0507 \newline
> \text{0.2} & 0.0642 \newline
> \text{0.5} & 0.0816 \newline
> \text{1.0} & 0.0865 \newline
> \hline
> \end{array}
>
> The results show that increasing the noise level of future states decreases action generation accuracy, confirming that the model effectively leverages the learned dependency between modalities to guide control.
>
> [1] Wang et al., Adaptive Stochastic Coefficients for Accelerating Diffusion Sampling, NeurIPS 2025
>
> ---
>
> ## [W3] Test-time scaling inference latency
>
> We give an additional analysis of sampling latency with higher denoising steps below. We clarify that the 0.104s latency mentioned in the paper was a conservative baseline measured on a slower GPU without any optimizations. Its primary purpose was to demonstrate that the dual-stream architecture does not impose a significant computational burden compared to standard single-stream action-DiT models. To more accurately reflect deployment conditions on a Franka Research 3, we re-benchmarked using an RTX 5090 GPU and TorchInductor compilation.
>
> \begin{array}{l|cc}
> \hline
> \text{Vision Denoising Steps} & \text{Latency (ms)} & \text{Control Frequency}  \newline
> \hline
> \text{4} & 24.59 & \sim 40 \text{Hz} \newline
> \text{16} & 72.64 & \sim 14 \text{Hz} \newline
> \text{32} & 127.14 & \sim 8 \text{Hz} \newline
> \text{64} & 246.06 & \sim 4 \text{Hz} \newline
> \hline
> \end{array}
>
> This allows us to utilize 4-step denoising with a control frequency of 40Hz, while scaling to 32-step denoising at around 8Hz and 64-step denoising at approximately 4Hz. We believe that more sophisticated optimization on the hardware and software side will lead to further latency reductions. However, as this is primarily an engineering challenge rather than a core theoretical concern, we relegate such optimizations to future work. Finally, we emphasize that test-time scaling is a tunable parameter. DUST with only 4 denoising steps already significantly outperforms baselines like GR00T-N1.5 and FLARE. This inherent flexibility allows DUST to maintain high-frequency feedback for dynamic grasping while reserving higher denoising steps for less time-sensitive tasks that require maximum predictive precision.

---

> > ### Author Rebuttal · Reviewer_R5Wh · 2026-04-02
> >
> > Thank the author for the thorough latency analysis provided in the rebuttal. The updated numbers on RTX 5090 with TorchInductor are helpful. That said, even with these optimizations, the 4-step baseline still runs at ~40Hz, and scaling to 32/64 steps drops to 8Hz/4Hz, which remains a practical concern for closed-loop control in dynamic scenarios.
> > Beyond the diffusion-based comparisons in Table 8, I notice that latency measurements for π0 and π0-FAST are missing, despite both being evaluated as performance baselines in Tables 1–3. Since π0-FAST specifically adopts action tokenization for fast inference, including its latency would provide a more complete picture of where DUST stands in the efficiency-performance trade-off landscape. More broadly, a comparison against AR-based VLAs that only decode action tokens at inference time (without the overhead of generating 64 vision tokens per forward pass) would strengthen the paper's practical positioning.

---

> > > ### Author Response · Authors · 2026-04-06
> > >
> > > Thank you for the response to our rebuttal. We answer your remaining concerns one-by-one in what follows.
> > >
> > > ---
> > >
> > > ### Practical deployment of DUST
> > >
> > > We believe that DUST is well-suited for practical deployment in real-world settings. Our test-time scaling offers an *elective* path for maximum precision. Meanwhile, the default 4-step configuration operates at 40Hz, which is 4 times faster than the typical dynamic closed-loop control which requires a 10 Hz threshold. Furthermore, it outperforms π0 and π0-FAST by 10-30 \% in various benchmarks (e.g., Tables 1-3), while maintaining more than 3 times faster latency. In addition, since DUST generates action chunks with a horizon of $k=16$ rather than single-step predictions, even our 4Hz (64-step) setting generates 64 individual action commands per second. While this involves open-loop execution within the action chunks, this is within the operational requirements of contemporary robots for most tasks. (For context, π0 typically utilizes inference every 0.5-0.8 seconds. [1]).
> > >
> > > ---
> > >
> > > ### Latency comparison
> > >
> > > Regarding the latency comparison, we further provide comparison with π0 series models and a representative autoregressive VLA model (OpenVLA [2]) below:
> > >
> > > \begin{array}{l|cc}
> > > \hline
> > > \text{Model} & \text{Latency (ms)} & \text{Control Frequency}  \newline
> > > \hline
> > > \text{DUST 4-step} & 24.59 & \sim 40 \text{Hz} \newline
> > > \text{DUST 16-step} & 72.64 & \sim 14 \text{Hz} \newline
> > > \text{DUST 32-step} & 127.14 & \sim 8 \text{Hz} \newline
> > > \text{DUST 64-step} & 246.06 & \sim 4 \text{Hz} \newline
> > > \hline
> > > \text{π0} & 77.32 & \sim 13 \text{Hz} \newline
> > > \text{π0.5} & 85.45 & \sim 12 \text{Hz} \newline
> > > \text{π0-FAST} & 313.80 & \sim 3 \text{Hz} \newline
> > > \text{OpenVLA} & 183.15 & \sim 5.5 \text{Hz} \newline
> > > \hline
> > > \end{array}
> > >
> > > As shown, our 4-step configuration achieves 3-10 times lower latency than baselines. While the diffusion-based π0 and π0.5 model show latencies comparable to our 16-step setting, the autoregressive π0-FAST and OpenVLA models have latency on par or slower than our 32-step and 64-step settings. Note that this efficiency stems from the lighter VLM backbone, which allows DUST to execute faster than π0 and π0.5. Also, AR methods iterate over a heavy VLM backbone to generate each token, whereas DUST generates actions over a relatively lighter diffusion head.
> > >
> > > We will incorporate the practical deployment and comparison with other VLA models more clearly in our final draft. If you have any further questions/concerns, please do not hesitate to let us know.
> > >
> > > ---
> > >
> > > [1] Black et al., π₀: A Vision-Language-Action Flow Model for General Robot Control (RSS 2025) \
> > > [2] Kim et al., OpenVLA: An Open-Source Vision-Language-Action Model (arXiv 2024)

---

### Official Review · Reviewer_GnoN · 2026-03-13

**Soundness:** 3
**Presentation:** 4
**Significance:** 3
**Originality:** 3
**Overall Recommendation:** 5
**Confidence:** 4

**Summary:**

The paper presents Dual-Stream Diffusion (DSD), a framework designed to balance visual foresight and action precision in robotic manipulation. The core technical contribution is the Decoupled Noise Step (DNS), which prevents the "shortcut" problem where the action stream ignores conditioning and relies solely on static VLM features. The model uses SigLIP for visual latent representation and demonstrates strong performance on the GR-1 benchmark.

**Compliance With Llm Reviewing Policy:**

Affirmed.

**Key Questions For Authors:**

see weakness

**Limitations:**

yes

**Strengths And Weaknesses:**

Strengths
Technical Soundness: The implementation of the Decoupled Noise Step (DNS) is a clever and effective solution to a common failure mode in diffusion-based policies. It ensures that the ActionDiT is truly conditioned on the visual stream rather than bypassing it.

Comprehensive Evaluation: The paper provides a detailed analysis of inference times and scaling behaviors, making it a very polished contribution to the field.

Major Concerns
Inference Latency vs. Test-time Scaling:

Table 8 indicates an inference time of 0.104s per action chunk using the default 4 denoising steps.

However, the Test-time Scaling experiments in Section 5.3 show that the model achieves its peak performance on the GR-1 benchmark only when visual denoising steps are increased to 32.

There is a lack of analysis regarding closed-loop physical performance under these high-latency conditions. For a real Franka robot setup using asynchronous sampling, what is the actual achieved control frequency at 32 or 64 steps?

I am concerned that this level of latency might lead to failures in high-precision or time-sensitive tasks, such as dynamic grasping, which rely on rapid real-time feedback.

Vision Encoder Comparisons:

The model currently relies on SigLIP for its latent state representation.

While SigLIP is a strong baseline, the paper would be significantly strengthened—and potentially "irrefutable"—if the authors included comparisons with other leading representations like DINO or V-JEPA2 to demonstrate the framework's robustness across different visual backbones.

---

> ### Author Rebuttal · Authors · 2026-03-31
>
> Dear reviewer GnoN,
>
> Thank you for your valuable comments and suggestions in reviewing our work. We address each of your questions and concerns individually as follows.
>
> ---
>
> ## [W1] Test-time scaling inference latency
>
> We give an additional analysis of asynchronous sampling latency with higher denoising steps below. We clarify that the 0.104s latency mentioned in the paper was a conservative baseline measured on a slower server-side GPU without hardware-specific optimizations. Its primary purpose was to demonstrate that the dual-stream architecture does not impose a significant computational burden compared to standard single-stream action-DiT models. To more accurately reflect deployment conditions on a Franka Research 3, we re-benchmarked using an RTX 5090 GPU and TorchInductor compilation.
>
> \begin{array}{l|cc}
> \hline
> \text{Vision Denoising Steps} & \text{Latency (ms)} & \text{Control Frequency}  \newline
> \hline
> \text{4} & 24.59 & \sim 40 \text{Hz} \newline
> \text{16} & 72.64 & \sim 14 \text{Hz} \newline
> \text{32} & 127.14 & \sim 8 \text{Hz} \newline
> \text{64} & 246.06 & \sim 4 \text{Hz} \newline
> \hline
> \end{array}
>
> This allows us to utilize 4-step denoising with a control frequency of 40Hz, while scaling to 32-step denoising at around 8Hz and 64-step denoising at approximately 4Hz. We believe that more sophisticated optimization on the hardware and software side will lead to further latency reductions. However, as this is primarily an engineering challenge rather than a core theoretical concern, we relegate such optimizations to future work. Finally, we emphasize that test-time scaling is a tunable parameter. DUST with only 4 denoising steps already significantly outperforms baselines like GR00T-N1.5 and FLARE. This inherent flexibility allows DUST to maintain high-frequency feedback for dynamic grasping while reserving higher denoising steps for less time-sensitive tasks that require maximum predictive precision.
>
> ---
>
> ## [W2] SigLIP-2 alternatives
>
> We believe that our framework can be robustly applied to any sufficiently powerful image embedding model, and provide additional experimental results on the RoboCasa dataset with DINOv2 and Flux.1 VAE as the future state embedding model.
>
> \begin{array}{l|ccc|c}
> \hline
> \text{Model} & \text{PnP} & \text{OP/CL} & \text{Other} & \text{Avg.} \newline
> \hline
> \text{GR00T-N1.5} & 0.215 & 0.603 & 0.468 & 0.417 \newline
> \text{FLARE} & 0.230 & 0.648 & 0.498 & 0.446 \newline
> \hline
> \text{DUST + SigLIP-2 (Default)} & 0.295 & 0.760 & 0.510 & 0.501 \newline
> \text{DUST + DINOv2} & 0.281 & 0.770 & 0.552 & 0.516 \newline
> \text{DUST + Flux.1 VAE} & 0.285 & 0.700 & 0.530 & 0.491 \newline
> \hline
> \end{array}
>
> The results in the table above demonstrate that the performance gains of DUST are not tied to a specific choice of visual representation, but rather stem from our dual-stream architecture and decoupled training objective.

---

> > ### Author Rebuttal · Reviewer_GnoN · 2026-04-04
> >
> > I have no further discussion. I will maintain my score 5, Good luck

---

### Decision · Program_Chairs · 2026-04-30

**Decision:**

Accept (regular)

**Comment:**

This paper was reviewed by three experts in the field and received final ratings of two Accept ratings and one Weak Accept. Based on the reviewers’ feedback, the decision is to recommend the paper for acceptance to ICML 2026. Reviewers viewed the paper positively, highlighting a technically sound and well-executed contribution with strong empirical support across simulation and real-world robotics settings. Reviewers emphasized the breadth of the evaluation, including RoboCasa, GR-1, real Franka experiments, transfer via action-free video pretraining, and heterogeneous joint training, and several noted that the paper is clearly presented and supported by meaningful ablations.

Overall, the paper is a strong contribution to robot learning and vision-language-action modeling. In the final version, I ask the authors to incorporate the relevant points from the discussion and rebuttal into the paper, as these clarifications would further strengthen the presentation and practical impact of the work. We congratulate the authors on the acceptance of their paper.